# Mitochondrial event localiser (MEL) to quantitativelydescribe fission, fusion and depolarisation in the three-dimensional space

**Rensu P. Theart**[1]*, **Jurgen Kriel**[2], **André du Toit**[2], **Ben Loos**[2], **Thomas R. Niesler**[1]

**1** Department of Electrical and Electronic Engineering, Stellenbosch University, Stellenbosch, Western Cape, South Africa, **2** Department of Physiological Sciences, Stellenbosch University, Stellenbosch, Western Cape, South Africa

* rptheart@sun.ac.za

**Data Availability Statement:** All the sample images that were analysed in the paper is available at: https://doi.org/10.6084/m9.figshare.13034129. v1 The mitochondrial event count results are

## Abstract

Mitochondrial fission and fusion play an important role not only in maintaining mitochondrial homeostasis but also in preserving overall cellular viability. However, quantitative analysis based on the three-dimensional localisation of these highly dynamic mitochondrial events in the cellular context has not yet been accomplished. Moreover, it remains largely uncertain where in the mitochondrial network depolarisation is most likely to occur. We present the mitochondrial event localiser (MEL), a method that allows high-throughput, automated and deterministic localisation and quantification of mitochondrial fission, fusion and depolarisation events in large three-dimensional microscopy time-lapse sequences. In addition, MEL calculates the number of mitochondrial structures as well as their combined and average volume for each image frame in the time-lapse sequence. The mitochondrial event locations can subsequently be visualised by superposition over the fluorescence micrograph z-stack. We apply MEL to both control samples as well as to cells before and after treatment with hydrogen peroxide ($H_2O_2$). An average of 9.3/7.2/2.3 fusion/fission/depolarisation events per cell were observed respectively for every 10 sec in the control cells. With peroxide treatment, the rate initially shifted toward fusion with and average of 15/6/3 events per cell, before returning to a new equilibrium not far from that of the control cells, with an average of 6.2/6.4/3.4 events per cell. These MEL results indicate that both pre-treatment and control cells maintain a fission/fusion equilibrium, and that depolarisation is higher in the post-treatment cells. When individually validating mitochondrial events detected with MEL, for a representative cell for the control and treated samples, the true-positive events were 47%/49%/14% respectively for fusion/fission/depolarisation events. We conclude that MEL is a viable method of quantitative mitochondrial event analysis.

## Introduction

Mitochondria are highly dynamic organelles that do not operate in a stagnant or isolated manner. Rather, they function in a highly energetic and networked fashion, continuously subjected

available here: https://doi.org/10.6084/m9.figshare.13034159.v1.

**Funding:** The authors received no specific funding for this work. This work was supported by the South African Medical Research Council (SAMRC), the Cancer Association of South Africa (CANSA), and Telkom South Africa and the South African National Research Foundation (NRF). Nvidia corporation, as a commercial funder, sponsored the GPU that was used in this project. The funders had no role in study design, data collection and analysis, decision to publish, or preparation of the manuscript.

**Competing interests:** Nvidia corporation sponsored the GPU that was used in this project. The authors of this paper have filed a patent for the method described in this paper under the name "Mitochondrial event localiser (MEL) to quantitatively describe fission, fusion and depolarisation in the three-dimensional space" (South African Provisional Application No. 2020/00654). There are no products in development or marketed products to declare. This does not alter our adherence to all the PLOS ONE policies on sharing data and materials. There are no further conflicts of interest associated with this publication.

to rapid remodelling events referred to as fission (fragmentation) and fusion [1]. These events serve as a critical quality control mechanism, not only to adapt and respond to changing metabolic demands but also to enable separation of damaged mitochondria from those in an interconnected state and thereby exposing them to specific degradation [2]. The elimination of damaged mitochondria, mediated through a mitochondrial specific autophagy referred to as mitophagy, decreases the risk of mitochondrial DNA (mtDNA) mutation accumulation. It also improves the electron transport chain (ETC) efficiency [3–6]. The transport of electrons between different complexes on the mitochondrial membrane results in the maintenance of a continuous voltage across the inner mitochondrial membrane (IMM) referred to as the mitochondrial membrane potential ($\Delta\psi$m) [7]. Mitochondrial dynamics are highly dependent on this membrane potential. Excessive membrane potential dissipation has, for example, been observed to render fragmented mitochondria incapable of re-fusing, thereby metabolically debilitating the cell [7–9]. Furthermore, the recruitment of the mitophagy initiation protein PTEN-induced kinase 1 (PINK1) only occurs once a certain depolarisation threshold is reached [10]. Failure to eliminate damaged mitochondria is a hallmark of neurodegenerative diseases such as Parkinson's disease [11], while certain heritable diseases, including Charcot-Marie-Tooth Type IIA, are related to dysregulated mitochondrial dynamics [12]. This underscores the important role mitochondrial fission and fusion play, not only in maintaining mitochondrial homeostasis, but also in preserving overall cellular viability [13]. Furthermore, there is much interest in quantitatively describing mitochondrial dynamics to better discern the role of mitochondrial dysfunction in pathology and pathogenesis of major human diseases and to determine whether changes in fission and fusion are adaptive or maladaptive. This includes the role of mitochondrial function in ageing [14, 15] in context of mitochondrial quality control and mitophagy [16] as well as in the context of mitochondrial directionality and network distribution [17]. However, a tool that assists in the rapid characterisation, localisation and quantification of mitochondrial fission, fusion and depolarisation events in the subcellular, three-dimensional space is currently not available. It is thus not surprising that the accurate quantification of these dynamic changes is attracting research attention is a lot of interest among researchers to quantify these dynamic changes accurately.

Previous attempts to better describe and unravel the interplay between mitochondrial dynamics and cell death onset have largely focused on either the change in the mitochondrial network of the entire cell or the rate at which fission and fusion occurs [11, 18, 19]. The accurate description of the quantitative relationship between fission and fusion dynamics in a three-dimensional cellular context, in order to detect deviation from its equilibrium, has however remained challenging. Moreover, although it is becoming increasingly clear that intracellular localisation of mitochondria is indicative of regional specific functions, for example the reliance of cellular organelles on mitochondrial ATP provision, it remains largely uncertain where in the mitochondrial network depolarisation is most likely to occur. It therefore remains to be determined which areas of the mitochondrial network are preferentially depolarised to facilitate either transportation or degradation and how these areas relate to the mitochondrial morphometric parameters usually employed. Currently, the methodologies available to address this question are limited, particularly in the context of three-dimensionally-based quantification of fission and fusion dynamics concomitantly with the onset of mitochondrial depolarisation. To gain a better understanding of mitochondrial dynamics, a high-throughput, automated and deterministic method that is able to localise and quantify the number of mitochondrial events in large three-dimensional (3D) time-lapse sample sets would therefore be advantageous. Such a method would allow subsequent quantitative description of the fission/fusion equilibrium as well as the extent of depolarisation.

Using time-lapse microscopy data of cells labelled for mitochondria, it has previously been observed that fission and fusion are rapid events occurring within a five second time frame under homeostatic conditions [20]. This work, however, did not investigate mitochondrial depolarisation. Current methods typically rely on the manual comparison of two time-lapse image frames in order to observe where fission, fusion or depolarisation occurred. This very labour intensive approach makes it challenging to gain comprehensive insights into the mitochondrial dynamics of a whole three-dimensional sample. It is usually unclear whether mitochondrial fission and fusion events are changing spatio-temporally, thereby shifting the equilibrium towards either fission or fusion. In this context, it is also often not clear whether a cell with a more extensively fused mitochondrial network is in transition or whether is has established a new equilibrium between fission and fusion by means of a newly adapted net contribution of fusion events. Similarly, a cell with greater mitochondrial fragmentation may establish a new equilibrium between fission and fusion as an adaptive response, in order to maintain and preserve this fragmented morphology. Hence, the micrograph reflects an underlying relationship between fission and fusion and this relationship is currently challenging to describe. Although mitochondrial photoactivation [18, 19] provides a highly selective tool to quantitatively assess mitochondrial dynamics, it does not reveal the relative contribution of fission and fusion events to the observed dynamics. For example, it has been noted that mitochondrial fusion is often enhanced during adaptations to metabolic perturbations, particularly in the perinuclear region, and may protect mitochondria from degradation [21]. On the other hand, a degree of fragmentation is required and desirable to allow, for example, mitochondrial transport in neurons to reach synaptic connections. Yet, describing the precise occurrence of fission, fusion and depolarisation has remained challenging.

To address these current challenges, we have developed an approach that automatically localises and count the number of mitochondrial events occurring between two micrograph frames in a time-lapse sequence. The results of this automatic analysis allow the quantitative assessment of fission, fusion and depolarisation localisation. This provides insight into the spatio-temporal change of mitochondrial events in a time-lapse sequence and can also be used to compare mitochondrial event dynamics under different treatment conditions. We refer to this newly developed procedure as the *mitochondrial event localiser* (MEL), because it allows us to indicate the precise three-dimensional location at which fission, fusion and depolarisation are likely to occur next. Moreover, MEL enables the determination of the individual locations of smaller structures that fuse to form a larger central structure in the two time-lapse frames that are considered. Similarly, the locations of smaller structures that will separate from a common central structure due to fission are identified. In doing so, MEL provides a platform to better understand mitochondrial dynamics in the context of health and disease, with both screening and diagnostics potential. This approach is, as far as we are aware, the first automated method for the detection of depolarised mitochondria in the context of fission and fusion events. MEL can serve both as a standalone method or as part of the broader mitochondrial analysis pipeline to enable high-throughput analysis of time-lapse data.

## Method

The development of MEL was inspired by previous work which applies a "vote casting" methodology to individual mitochondrial structures in two consecutive time-lapse frames in order to identify the likely mitochondrial event (fission, fusion, or none) the structure will undergo [20]. The purpose of this approach was specifically to count the number of fission and fusion events and can not easily be extended to determine their location. MEL was designed from the outset to both count the number of mitochondrial events and to localise each in 3D space.

The MEL algorithm processes a fluorescence microscopy time-lapse sequence of z-stack images that were stained for mitochondria and produces 3D locations indicating where mitochondrial events are likely to occur at each time step. These locations can subsequently be superimposed on the z-stacks in order to indicate the different mitochondrial events. The algorithm has been organised into two consecutive steps, namely the image pre-processing step which normalises and prepares the time-lapse frames, and the automatic image analysis step which calculates the location of the mitochondrial events based on the normalised frames.

Since some of the detected events could be false-positives as a consequence of the thresholding step used to join two closely-spaced but still separate mitochondrial structures, MEL also allows each event to be subsequently validated and removed from the visualisation and event counts.

The code that implements MEL is available for download at https://github.com/rensutheart/MEL.

## Image pre-processing

The image pre-processing step receives a time-lapse sequence of z-stacks as input and begins by selecting two z-stacks, which we will refer to as Frame 1 and Frame 2, for further processing. Depending on the temporal resolution that is desired, the selected z-stacks can either be consecutive time-lapse frames, or some number $k$ of intermediate frames may be skipped. Since mitochondrial movement is highly dynamic, a time interval that is too long might prevent the algorithm from accurately matching mitochondrial structures between the chosen images frames, thereby degrading its ability to determine mitochondrial events. The required time interval for accurate results is dependent on the motility of the mitochondria. In our experiments, time intervals of 10-30 seconds produced favourable results. Shorter time intervals may also be used [20]. The selected Frames 1 and 2 are then each processed in the same way by the image pre-processing step to generate several new image stacks which are passed to the automatic image analysis step. The image pre-processing step, which will be described in the remainder of this section, is illustrated in Fig 1.

Since MEL is not based on the analysis of a single cell, it is not necessary to select regions of interests (ROIs) before the analysis. However, since the image acquisition parameters that are used vary widely between different time-lapse sequences, we first normalise the fluorescence intensity data of the z-stacks. This is necessary since MEL relies on an accurate binarisation of the fluorescence image stack to identify the voxels that contain mitochondria. These thresholding algorithms require images to be sharp, contain minimal noise, and have good contrast between the foreground and background. The normalisation and binarisation, which form the initial part of the MEL image pre-processing step, were guided by previous work focusing on quantitative mitochondrial network analysis [20, 22–24].

The first step in normalising the raw micrographs is to apply deconvolution to the z-stacks using a point spread function (PSF) that is estimated from the microscope's acquisition parameters. Next, the resolution of the micrographs constituting the z-stack is increased by a factor of 1.5 using bilinear interpolation. This reduces the possibility that two adjacent but unconnected structures are erroneously joined after binarisation. In general, larger scaling factors of two or three should further improve this. We were, however, unable to use these scaling factors for our analysis due to RAM limitations of the computer system on which the analysis was performed. This is followed the application of Gaussian blur ($\sigma_{2D} = 1$) to the micrographs which reduces noise before thresholding. Next, we apply contrast stretching to the z-stack, with a saturation of 0.3% of the voxels (the default value in Fiji), to normalise the fluorescence intensity between micrographs.

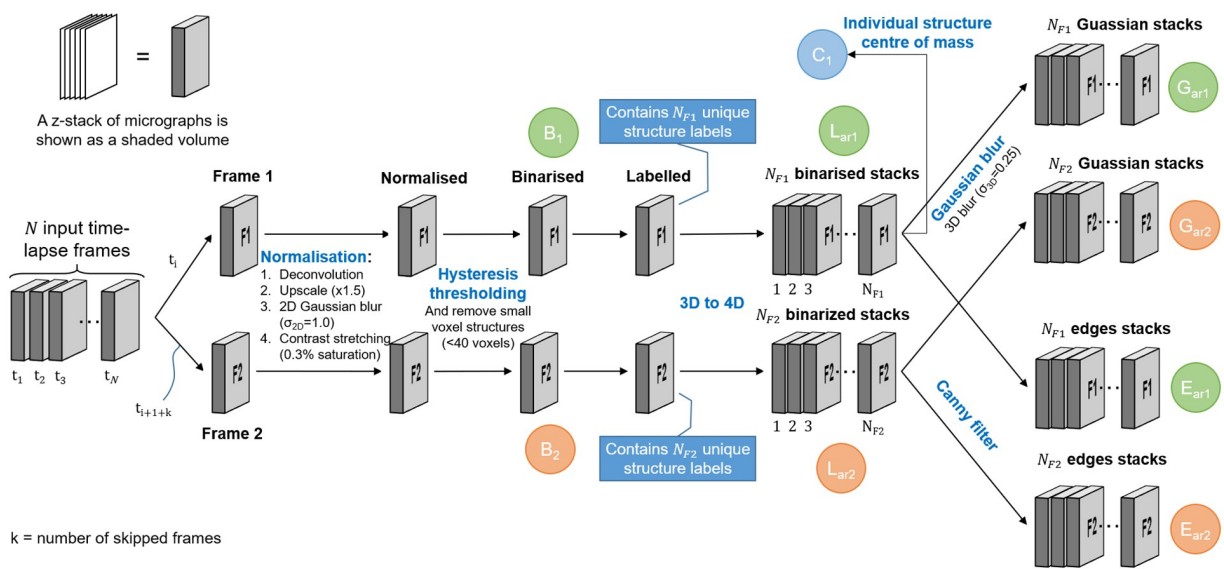

**Fig 1. Image pre-processing begins by choosing two frames from the input time-lapse sequence, Frame 1 and Frame 2.** Normalisation, binarisation, and labelling are then performed on both frames. The labelled frames are each separated into an array of z-stacks, where each stack contains only a single labelled structure. Each of the stacks in the array of binarised stacks, for both Frames 1 and 2, is then Gaussian blurred and Canny filtered. The latter contains only the edges of the labelled structures. The stacks that are used during the automatic image analysis step are labelled in green for Frame 1 and orange for Frame 2. Finally, an array containing the centre of mass for each labelled structure is shown in blue.

The normalised frames are binarised by applying adaptive hysteresis thresholding to the z-stack, although other thresholding methods might also be suitable. Hysteresis thresholding is very effective in removing background voxels that have intensities similar to the mitochondrial object voxels. It uses two threshold values. Voxels that have an intensity above the high threshold are considered to belong to the object and voxels that have an intensity below the low threshold are considered to belong to the background. Voxels that have an intensity between the low and high thresholds are considered to belong to the object if they are connected to other object voxels [25]. We automatically calculated the low threshold at the edge of the histogram valley of the background voxel intensities, and the high threshold at the halfway point between the low threshold and the maximum intensity. This is illustrated in S1 Fig.

Since it has been observed that some noise remnants were also being binarised, we removed any structure containing less than 40 voxels in the upscaled images. We determined this value empirically by considering the average size of the binarised noise structures. In concrete terms, this is equivalent to a circle with a diameter of 5 pixels in the original image, or in our case a circular structure with a physical diameter of about 0.6 $\mu$m. This z-stack that results from this is labelled $B$ in Fig 1. Note that we differentiate between the stacks of Frame 1 and Frame 2 in Fig 1 by using subscripts 1 and 2, respectively.

After binarisation, we assign a label to each separate 3D voxel structure. The total number of labels in Frame 1 and Frame 2, as depicted in Fig 1, are $N_{F1}$ and $N_{F2}$, respectively. Later in the automatic image analysis step, each labelled structure in each frame will be compared with labelled structures both within the same frame as well as in the other frame. For this reason, we separate the labelled z-stack into an array of z-stacks, where the $i$th stack in the array contains only the binarised voxel structure associated with label number $i$. We refer to this array of stacks as $L_{ar}$. The subscript "$ar$" is used to indicate an *array* of stacks, containing $N_{F1}$ and $N_{F2}$

stacks for Frame 1 and Frame 2, respectively. This step can be thought of as adding the structure label as a fourth dimension to the data and later enables fast comparison between different labelled structures.

From the array of labelled z-stacks $L_{ar}$ we create two similar arrays also required by the automatic image analysis step. The first is the result of applying a 3D Gaussian filter ($\sigma_{3D} = 0.25$) to each z-stack in the array. This slightly blurs the edges of the labelled stacks and results in the array of z-stacks $G_{ar}$. The Gaussian blurring enhances the ability of the algorithm to match the moving mitochondrial structures between two frames by slightly inflating the structures. The second array of stacks is generated by removing all voxels that are not located on the edges of the 3D labelled structures in $L_{ar}$ by using Canny edge detection [25]. We refer to the resulting array of z-stacks as $E_{ar}$. This is performed mainly to improve the efficiency of the algorithm that later determines the location of fission and fusion events.

This concludes the image pre-processing step. The tuneable parameters in this step is the volume of the structures that are considered as noise, and the standard deviation with which the z-stacks are Gaussian blurred. The volume should be determined empirically by considering the average size of the binarised noise without erroneously removing true mitochondrial structures. From our testing, the value of $\sigma_{2D}$ in the normalisation step should take values between 0.5 and 1.5 (0.04-0.12 $\mu$m). $\sigma_{3D}$ in the 3D Gaussian filter step should take values between 0.2 and 0.5 (0.02-0.04 $\mu$m in x-y and 0.1-0.25$\mu$m in z) due to the limited resolution in the z-dimension. These values are, however, subject to different image acquisition parameters. The z-stack $B$, and the arrays of z-stacks $G_{ar}$ and $E_{ar}$ are now passed into the automatic image analysis step that calculates the location of the mitochondrial events.

## Automatic image analysis step

The purpose of the automatic image analysis step is to generate a list of locations in Frame 1 at which the mitochondrial events occur. It achieves this by receiving the z-stacks that were generated in the image pre-processing step. These are then automatically analysed to produce a list of the mitochondrial event locations. The automatic image analysis step is shown in Fig 2.

**Mitochondrial events.** The three different mitochondrial events that we consider, namely fission, fusion and depolarisation, are defined in terms of changes in mitochondrial morphology that occur in the time interval between Frames 1 and 2. Specifically, fission is defined as the separation of a larger mitochondrial structure to two smaller structures. Similarly, fusion is defined as the joining of two mitochondrial structures to form a single larger structure. Finally, depolarisation is defined as the disappearance of a mitochondrial structure from Frame 1 to Frame 2 manifested by a complete loss of fluorescence signal. If more than two structures fuse to a single central structure, or if one structure undergoes fission to form more than two smaller structures, then separate locations are assigned to each.

**Process.** In order to determine the location of mitochondrial events, we first identify which of the voxel structures labelled during the image pre-processing step are candidates for fission and fusion. First, the voxel structures in Frame 1 that occupy the same space as a voxel structures in Frame 2, and vice versa, are identified. For the sake of conciseness, we will refer to such colocation of structures between the two frames as overlap. Using this overlap information, we can determine which voxel structures are associated with each other within the same frame. In Frame 1, these associated structures will be the fusion candidates, and in Frame 2 they will be the fission candidates. Structures in Frame 1 that have no associated structures in Frame 2 undergo depolarisation.

To determine the associated structures between Frame 1 and Frame 2, we first calculate a matrix $V$ containing the overlapping volume (in voxels) of each structure in Frame 1 with each

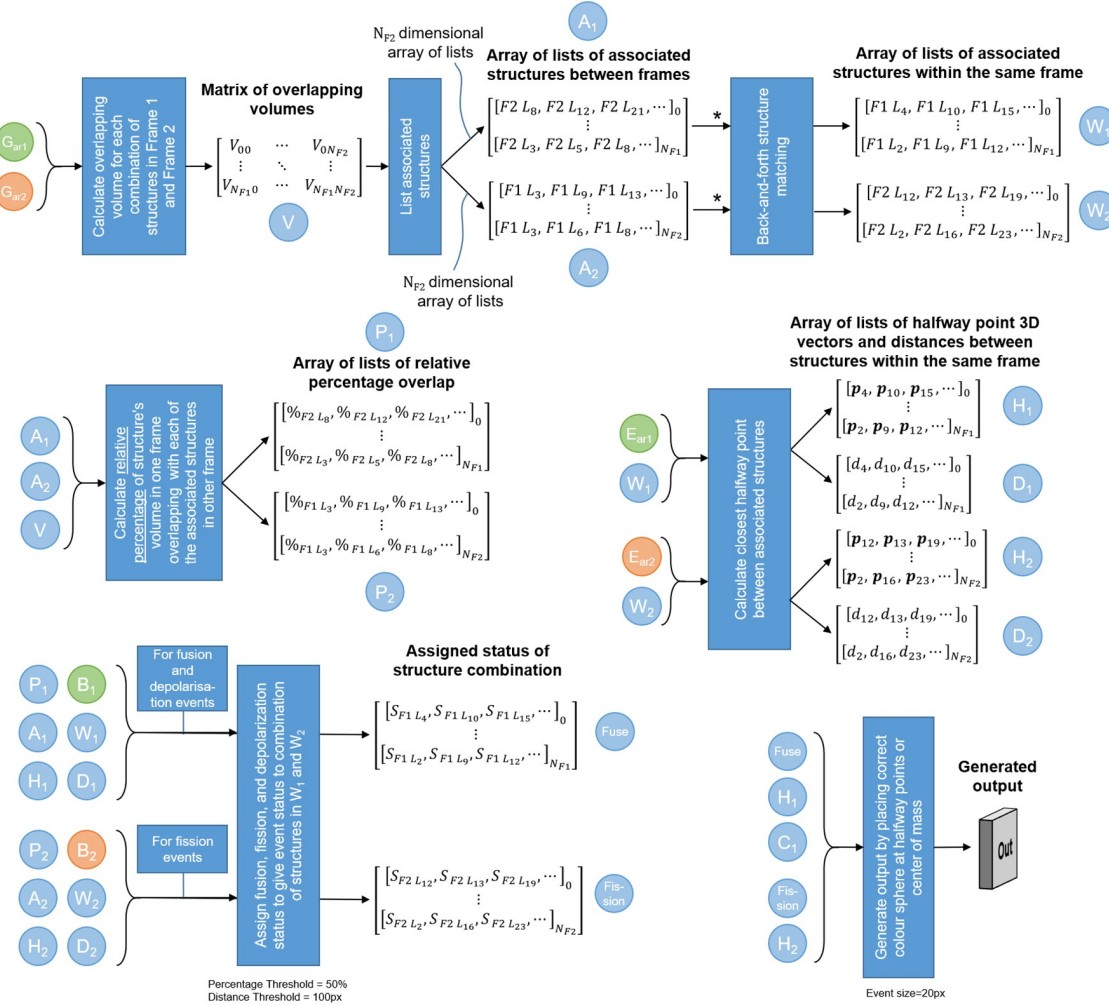

**Fig 2. Automatic image analysis begins by calculating the overlapping volume of each structure in Frame 1 with each structure in Frame 2.** $V_{xy}$ refers to the number of voxels that overlap between structure $x$ in Frame 1 and structure $y$ in Frame 2. Where $x = 0$ or $y = 0$ refers to the background. Using matrix $V$, the labelled structures that are associated with each other both between Frames 1 and 2, as well as within the same frame, can be calculated. The relationships are encoded as arrays of lists, where each list is related to the labelled structure corresponding to the array index. Therefore, the 8$^{\text{th}}$ list in the array relates to labelled structure number 8. $F2L_8$ refers to label number 8 in Frame 2 that is associated with the structure at the array index. Next, the relative percentage of overlap (e.g. $\%_{F2L8}$) is calculated for the associated structures between the frames, and is interpreted in conjunction with $A_1$ and $A_2$. The midway points (e.g. $\mathbf{p}_4$) and distances (e.g. $d_4$) between two structures within the same frame are calculated, and are interpreted in conjunction with $W_1$ and $W_2$. Using these arrays of lists, the status (e.g. $S_{F1L_4}$) of fission, fusion, depolarisation or no event can be determined for each mitochondrial structure. Finally, the event can be overlaid on the original z-stack at the location of the midway points.

structure in Frame 2 using the blurred arrays of z-stacks $G_{ar_1}$ and $G_{ar_2}$. Label number 0 is reserved for the background. Blurred stacks are used in order to allow for movement of the mitochondria between frames. Using matrix $V$, we calculate two arrays $A_1$ and $A_2$. Each entry in array $A_1$ is a list of all the structures in Frame 2 that overlap with a particular structure in Frame 1. Similarly, $A_2$ indicates the structures in Frame 1 that overlap with a particular structure in Frame 2.

**A: Fusion candidate determination**

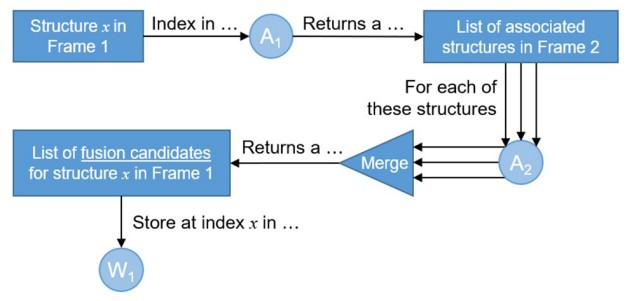

**B: Fission candidate determination**

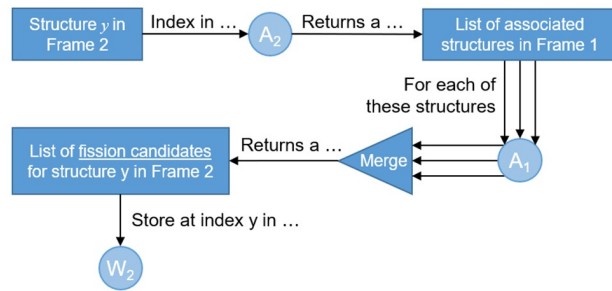

**Fig 3. The back-and-forth structure matching algorithm used to determine mitochondrial structures that are candidates for fission and fusion.**

The information in $A_1$ and $A_2$ allow us to determine which structures in Frame 1 are fusion candidates, and which structures in Frame 2 are fission candidates using an algorithm we call *back-and-forth structure matching*. For each structure in Frame 1, $A_1$ is used to determine all associated structures in Frame 2. For each of these associated structures in Frame 2, $A_2$ is used to determine all the associated structures in Frame 1. This list of Frame 1 structures comprises the fusion candidates for the Frame 1 structure being considered. An identical but opposite procedure is used to determine the fission candidates in Frame 2. The resulting two arrays of lists are denoted $W_1$ and $W_2$. In the remainder of automatic image analysis step we will determine which of these candidates truly underwent the mitochondrial event. The back-and-forth structure matching algorithm is illustrated in Fig 3 and is practically demonstrated with the help of a synthetic example in S1 Appendix.

Many of the mitochondrial event candidates identified in the previous step might be a result of coincidental structure overlap. These overlaps could either be due to mitochondrial movement in the time interval between Frame 1 and Frame 2, or as a result of comparing the blurred frames $G_{ar_1}$ and $G_{ar_2}$. Blurring was introduced to compensate for mitochondrial movement, but has the consequence of increasing the number of false overlaps. To remove these coincidental overlaps, we begin by calculating the relative percentage overlap between associated structures in Frame 1 and Frame 2. For each structure in Frame 1, the set of associated structures in Frame 2 is identified using $A_1$. Then, using $V$, the overlapping volume of each of these associated structures is normalised relative to the combined overlapping volume of all the associated structures. This results in the relative percentage overlap that each structure in Frame 2 has with the structure in Frame 1. We will later use these percentage to determine the

likely candidate structures to undergo fission or fusion. These overlap percentages are stored in an array of lists $P_1$, which has the same structure as $A_1$. An analogous procedure is followed for all the structures in Frame 2, in this case using $A_2$, resulting in $P_2$. $P_1$ and $P_2$ are used to remove mitochondrial event candidates whose proportion of volume overlap is small.

The back-and-forth structure matching algorithm can also produce false matches. For example, when two small structures are on opposite sides of a larger central structure in Frame 1 and both fuse with this central structure in Frame 2, the smaller structures in Frame 1 will be identified as being associated with each other. This is, however, a false match since no mitochondrial event occurs between them. Such false matches can easily be avoided by detecting the presence of a third mitochondrial structure between the two candidates for association, or by determining that the candidate structures are too far apart for a mitochondrial event to be feasible. We therefore calculate the shortest vector length as well as the point in 3D space midway between all structure pairs identified in $W_1$ and $W_2$. This midway point represents a likely location of fission and fusion events. The shortest vector lengths can be computed from the information in $E_{ar_1}$ and $E_{ar_2}$. The resulting list of shortest distances are denoted by $D_1$ and $D_2$ and the midway points by $M_1$ and $M_2$ for fusion and fission respectively. Note that the length of each array in the lists matches those of $W_1$ and $W_2$ for the same subscript.

Using $A_1$ and $A_2$, we can find the structures in Frame 1 that have no associated structures in Frame 2, indicating that they will depolarise. Next, the false fission and fusion candidate structures in $W_1$ and $W_2$ can be filtered out by using $D_1$, $D_2$, $M_1$ and $M_2$ to determine the final sets of fission and fusion events. First, a distance threshold corresponding to the maximum distance that a mitochondrial structure could be expected to move is determined. Then, all candidates in $W_1$ and $W_2$ for which the respective distances in $D_1$ and $D_2$ exceed this threshold are removed. Next, fission and fusion candidates separated by a third structure are removed by considering the binarised stacks $B_1$ and $B_2$ in conjunction with the midway points in $M_1$ and $M_2$. Finally, by using $P_1$, we consider two structures within Frame 1 that are both associated with the same structure in Frame 2 with a high relative percentage overlap as a fusion event. Similarly, using $P_2$, we consider two structures within Frame 2 that are both associated with the same structure in Frame 1 with a high relative percentage overlap as a fission event. Any structure pairs with a low and high, or a low and low, combination of relative percentage overlap is ignored. Empirically we have observed that in most cases the relative percentage overlap is either above 90% or below 10%. Therefore, we consider any percentage above 50% to be high. This step results in two more arrays of lists. The *Fusion* array indicates which structures in Frame 1 will fuse, depolarise, or undergo no event. The *Fission* array indicates which structures in Frame 2 are the result of a fission event.

The final step is to generate an output image by superimposing a colour label, in three-dimensional space, on the z-stack to indicate the location of each mitochondrial event at the midway point between the two structures that fuse in Frame 1, or using the midway point between the two structures that are the result of fission in Frame 2 as the location at which a structure in Frame 1 will undergo fission. The structures that will depolarise are indicated by placing a colour label at its centre of mass, which is calculated from the z-stack for the structure in question from the $L_{ar_1}$ array.

The automatic image analysis step, indicating which stacks and list of arrays are used for which parts of the algorithm, is shown in Fig 2. The tuneable parameters in this step is the percentage that is the threshold between a low and high relative overlap percentages and the distance threshold between structures that can be considered to be related to each other. In order to further clarify each step in the MEL procedure, we have included its application to a simple synthetic sample as a supplementary section.

### Biological sample investigation

Using a mammalian cell model, we apply and validate MEL by assessing the mitochondrial network under physiological as well as disrupted conditions. For this purpose we analysed control cells and contrasted these with cells imaged both before and after treated with hydrogen peroxide ($H_2O_2$) (Pinnacle Pharmaceuticals) to induce mitochondrial dysfunction [19, 26].

**Cell line maintenance.**    U-118MG cells were purchased from the American Type Culture Collection (ATCC) and supplemented with Dulbecco's Modified Eagles Medium (DMEM), 1% penicillin/streptomycin (PenStrep) (Life Technologies, 41965062 and 15140122) and 10% foetal bovine serum (FBS) (Scientific Group, BC/50615-HI) and incubated in a humidified incubator (SL SHEL LAB $CO_2$ Humidified Incubator) in the presence of 5% $CO_2$ at 37˚C.

**Microscopy.**    Live cell confocal microscopy of mitochondrial fission and fusion events was conducted using a Carl Zeiss Confocal Elyra PS1 microscope coupled to LSM 780 technology. Image stacks were acquired with a 0.5 $\mu$m step width using a Plan-Apochromat 100x/1.46 oil DIC M27 objective and a 561 nm laser as an illumination source together with a GaAsP detector. Prior to imaging, U-118MG cells were seeded in Nunc® Lab-Tek® II 8 chamber dishes and incubated with 100 nM tetramethylrhodamine-ethyl ester to allow for visualisation of the mitochondrial network (TMRE, Sigma Aldrich, 87917) for 5 minutes in the presence of 5% $CO_2$ at 37˚C. A baseline time series of mitochondrial dynamics was established by acquiring micrographs at 10 sec intervals for 30 cycles. In order to perturb the mitochondrial network, cells were treated with 500 $\mu$M $H_2O_2$ and images were acquired at 10 second intervals for 30 cycles.

Low laser power was used to ensure that photo bleaching and photooxidative stress was limited during this acquisition period. Since the total acquisition time for the treated cells was 10 minutes (5 minutes before treatment and 5 minutes after treatment), the control cells were also acquired for a total of 10 minutes in order to ensure that photo bleaching was limited (as shown in S2 Fig) and therefore the effects of the treatment on the cell could be interpreted as resulting primarily from the treatment itself. Only the data of the first 5 minutes of the control cells are shown in the results section.

TMRE was used due to its wide use in assessing mitochondrial dysfunction, and its exceptionally good signal-to-noise ratio. However, MEL would equally operate on image data acquired using other fluorescent probes such as MitoTrackers.

Care should be taken to not perturb the cellular microenvironment, due to, for example, the solvent used, T or pH fluctuations, since the mitochondrial network will be impacted.

## Results

The ability to automatically locate and count the number of mitochondrial events that occur in a sample is key to the investigation of mitochondrial dynamics. In this section, we assess and validate MEL by applying it to time-lapse sequences of mammalian cells taken before and after treatment with hydrogen peroxide ($H_2O_2$) to induce mitochondrial dysfunction, as well as under control conditions.

### Validation of MEL accuracy

MEL produces the locations at which mitochondrial fission, fusion and depolarisation events are likely to occur. Some of these events could, however, be false-positives either due to inaccuracies in the binarisation of the mitochondria or due to mitochondrial motility which prevents MEL from accurately matching structures between frames. In order to validate the events, a tool was developed that allows each event to be evaluated individually by a human expert

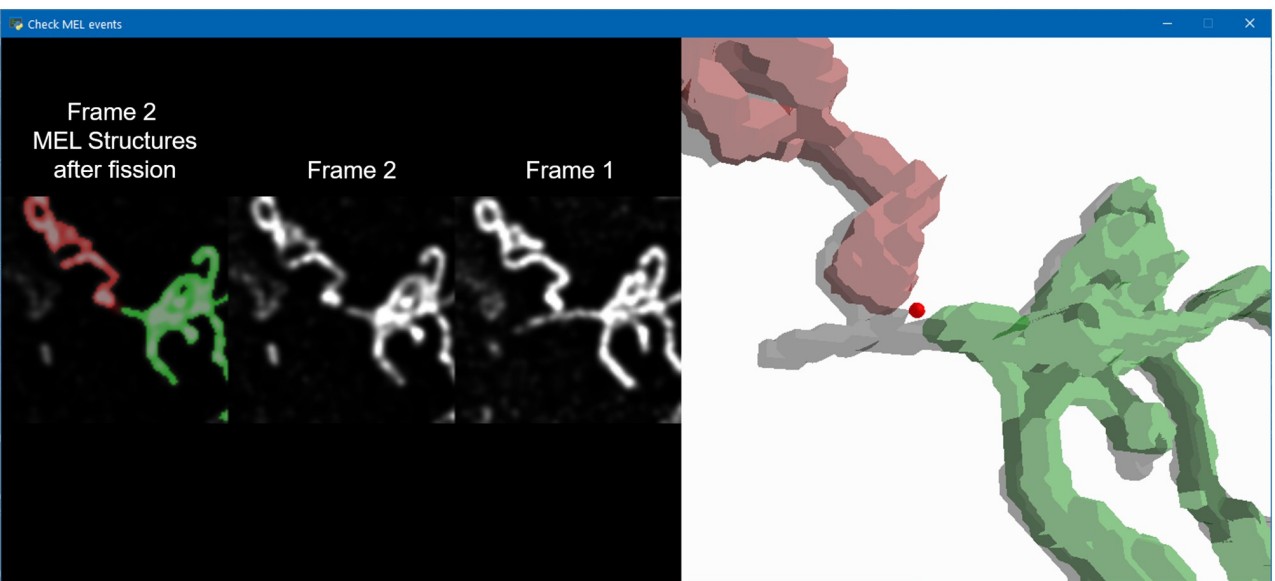

**Fig 4. The mitochondrial event validation tool can be used to identify and remove false-positive events that were detected by MEL.** In the left panel, the event-centred cropped normalised z-stacks for Frame 1 and Frame 2 are shown. The two binarised mitochondrial structures that will fuse are highlighted in red and green. In the case of fusion, these structures are superimposed on Frame 1. In the case of fission, as shown here, the structures are superimposed on Frame 2. The investigator can cycle through the different micrographs comprising the z-stack. In the right panel, three-dimensional reconstructions of the green and red mitochondrial structures are shown with the reconstructed structure into which the first two will fuse, or from which they underwent fission.

(Fig 4). This is helpful to enhance the accuracy of MEL for single-cell analysis since it produces a better estimation of the absolute number of mitochondrial events. In the left panel of Fig 4, a cropped view of the surrounding mitochondrial structures at the detected event location is shown. The binarised structures that have been classified to undergo fusion and the structures that would be produced as a result of fission are overlaid in red and green on Frame 1 in the case of fusion and Frame 2 in the case of fission (as shown in Fig 4). Frame 1 and Frame 2 are also shown without the binarised overlay to assist in the validation of the event. Since Frame 1 and Frame 2 are z-stacks, the investigator can cycle through the cropped view of the different micrographs in the z-stack. In the right panel, a three-dimensional reconstruction of the binarised structures in one frame is overlaid in red and green on a grey structure which is the binarised structure in the other frame that the green and red structures will fuse into or from which fission will occur. This three-dimensional view allows the investigator to navigate through the larger mitochondrial structures in question to observe the event in its three-dimensional context.

By using this validation tool, we determined that there were some scenarios in which MEL produced false-positive events. These scenarios are illustrated in Fig 5. Fusion and fission are sometimes falsely detected when two mitochondrial structures are in close proximity. This leads to a low-intensity bridge between the two structures. During binarisation, these structures are erroneously joined. Furthermore, depolarisation can sometimes be falsely detected when small mitochondrial structures move so far between Frame 1 and Frame 2 that MEL no longer considers them to be associated. Finally, occasionally two mitochondrial structures will fuse with a third structure in such a way that MEL fails to remove the event from the list of candidate events due to the third structure not being detected in between the two structures or

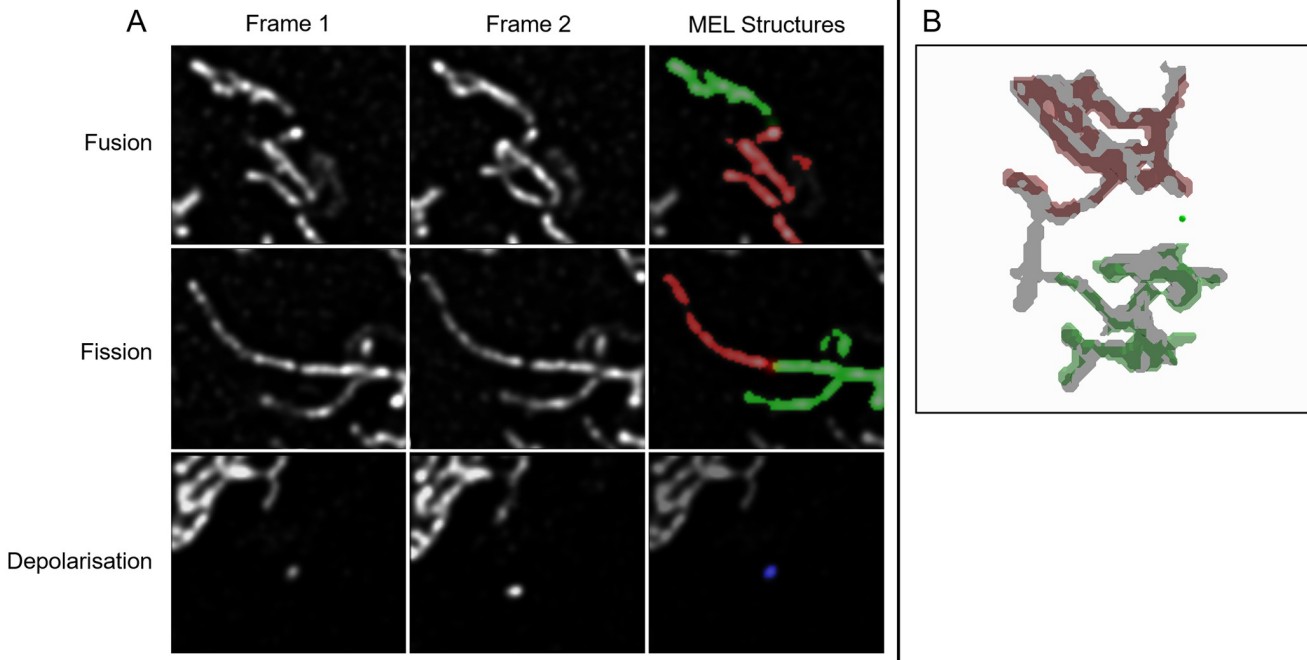

**Fig 5. Common scenarios in which false-positive events are detected.** A: An example illustrating how fission and fusion events can be incorrectly detected when two mitochondrial structures are too close together, causing a low-intensity bridge to be formed between them. This signal is then erroneously binarised as a single larger mitochondrial structure. Depolarisation can be detected incorrectly when a small structure is not matched correctly between Frames 1 and 2 as a result of mitochondrial motility. B: An example of two mitochondrial structures that fuse with a third in-between structure, but are erroneously considered by MEL to fuse with each other.

because the distance between the first two structures are smaller than the chosen distance threshold.

Next, we validated all the events detected by MEL in a representative time-lapse sequence of a cell in the control group as well as before and after $H_2O_2$ treatment. The results are summarised in Fig 6 and Tables 1 and 2. Even though the number of true-positive events is consistently smaller than the number of events detected by MEL, from Fig 6 it is clear that the overall trends throughout the time-lapse sequence as well as the fission:fusion ratio remain largely the same. Furthermore, when comparing different groups with one another, the relative differences between them have also remained the same. One advantage of validating the detected events is that that the variance in the data can be reduced. Further considerations when using MEL are discussed in S2 Appendix.

## Single-cell time-lapse analysis

MEL, coupled with the manual event validation, can be used for single-cell analysis to gain an understanding into the mitochondrial dynamics throughout a time-lapse sequence. In this section, we show a detailed visual analysis of representative cells from the control group as well as before and after $H_2O_2$ treatment.

Figs 7–9A show the maximum intensity projection (MIP) of the MEL output for every seventh frame in the time-lapse sequence. Column B shows the MIP of a region of interest (ROI), indicated by a white square in column A, for all the frames in the time-lapse sequence. The ROI was specifically chosen to highlight the detection and visualisation of mitochondrial

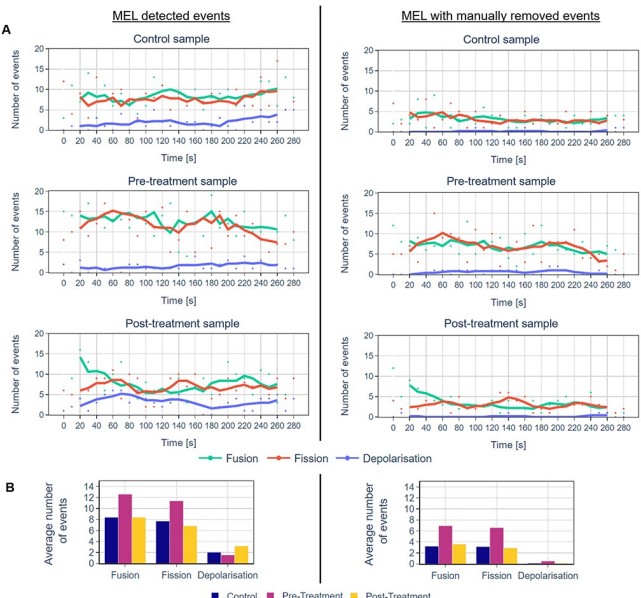

**Fig 6. Comparison of the number of mitochondrial events for representative cells from the control, before and after H₂O₂ treatment groups as detected by MEL and after erroneously detected events were manually removed.** The lines represent the five-frame simple moving average. A: A comparison of the number of events that were detected throughout the time-lapse sequence for the different groups. B: A comparison of the mean of the number of events for the control and treatment groups.

events. This column shows Frames 1 and 2 as used by MEL as well as the generated output frame. Column C shows a selection of the MEL ROI frames in three dimensions. Column D shows a graphical summary of some of the data extracted by MEL for each frame pair. This includes the five-frame simple moving average (SMA) of the number of mitochondrial events that occur, the total number of mitochondrial structures, as well as the average volume of these structures in voxels.

**Table 1. Comparison of the mean and standard deviation of the number of mitochondrial events for representative cells from the control, before and after H₂O₂ treatment groups as was detected by MEL and after erroneously detected events were manually removed.**

|  | Event type | MEL detected events | | MEL with manually removed events | |
|---|---|---|---|---|---|
|  |  | Mean | SD | Mean | SD |
| Control | Fusion | 8.4 | 3 | 3.2 | 1.9 |
|  | Fission | 7.7 | 3.5 | 3.1 | 2.1 |
|  | Depolarisation | 2.0 | 1.8 | 0.1 | 0.4 |
| Before | Fusion | 12.6 | 4.4 | 6.9 | 2.7 |
|  | Fission | 11.4 | 4.9 | 6.6 | 3.4 |
|  | Depolarisation | 1.6 | 1.1 | 0.5 | 0.6 |
| After | Fusion | 8.4 | 4.6 | 3.6 | 2.7 |
|  | Fission | 6.8 | 2.5 | 2.9 | 1.7 |
|  | Depolarisation | 3.1 | 1.8 | 0.1 | 0.4 |

The results are visualised in Fig 6.

**Table 2. Comparison of the total number of mitochondrial events for representative cells from the control, before and after H$_2$O$_2$ treatment groups as was detected by MEL and the number of true-positive events after erroneously detected events were manually removed.**

|  | Event type | MEL detected events | True positives | Accuracy |
|---|---|---|---|---|
| Control | Fusion | 243 | 93 | 38% |
|  | Fission | 224 | 92 | 41% |
|  | Depolarisation | 61 | 6 | 10% |
| Before | Fusion | 365 | 201 | 55% |
|  | Fission | 331 | 192 | 58% |
|  | Depolarisation | 47 | 17 | 36% |
| After | Fusion | 243 | 104 | 43% |
|  | Fission | 199 | 85 | 43% |
|  | Depolarisation | 94 | 6 | 6% |
| Total | Fusion | 851 | 398 | 47% |
|  | Fission | 754 | 369 | 49% |
|  | Depolarisation | 202 | 29 | 14% |

## Statistical comparison of different cell groups

In this section, we use MEL in the context of multiple single-cell analysis to determine whether specific mitochondrial events can be used to characterise cells under various treatment conditions.

For this analysis, we considered time-lapse sequences of four independent biological repeats of control cells and cells before H$_2$O$_2$ treatment and the same cells after H$_2$O$_2$ treatment. For each biological repeat, newly passaged cells were seeded and subsequently imaged. Since we

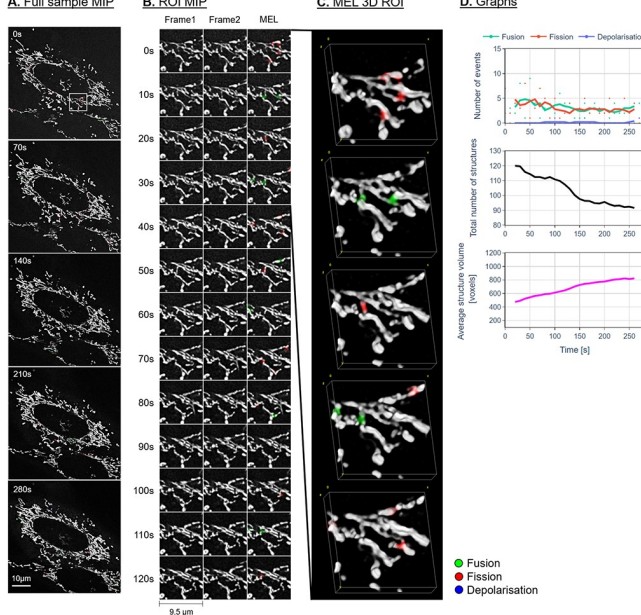

**Fig 7. Representative control sample.** A: Maximum intensity projection (MIP) of the MEL events overlaid on the entire sample image of every seventh frame in the time-lapse sequence. B: A region of interest (ROI) selection, indicated by the white square in column A. Frame 1 matches the time indicated, Frame 2 shows the subsequent time step, MEL shows the detected mitochondrial events overlaid on Frame 1. C: A selection of the ROI frames in column B, visualised in 3D using volume rendering. D: The five-frame simple moving average of the values calculated by MEL.

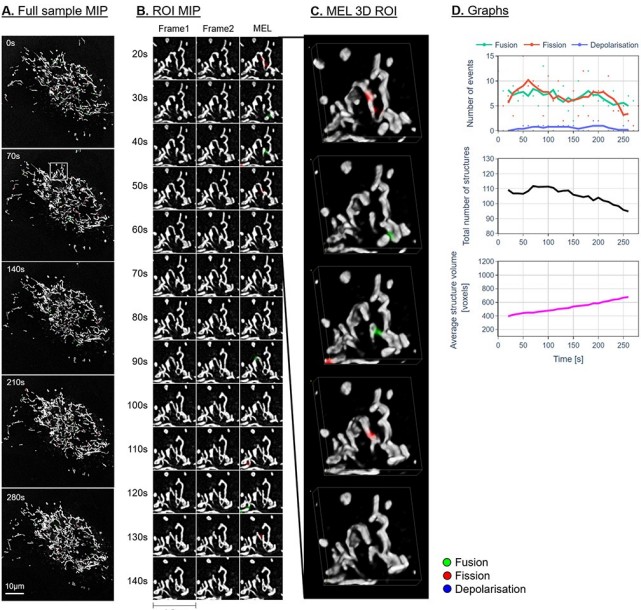

**Fig 8. Representative sample before H$_2$O$_2$ treatment.** Description as for Fig 7.

utilised 8-chamber dishes, with each independent experiment, also technical repeats were acquired.

The results are summarised in Fig 10 and in Table 3. Fig 10A show the average number of events that were detected by MEL at each time interval for all the samples in the different

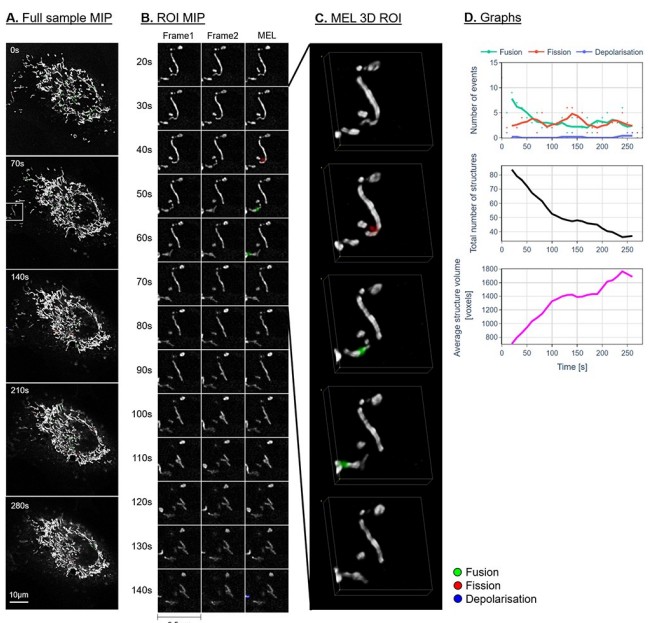

**Fig 9. Representative sample after H$_2$O$_2$ treatment.** Description as for Fig 7.

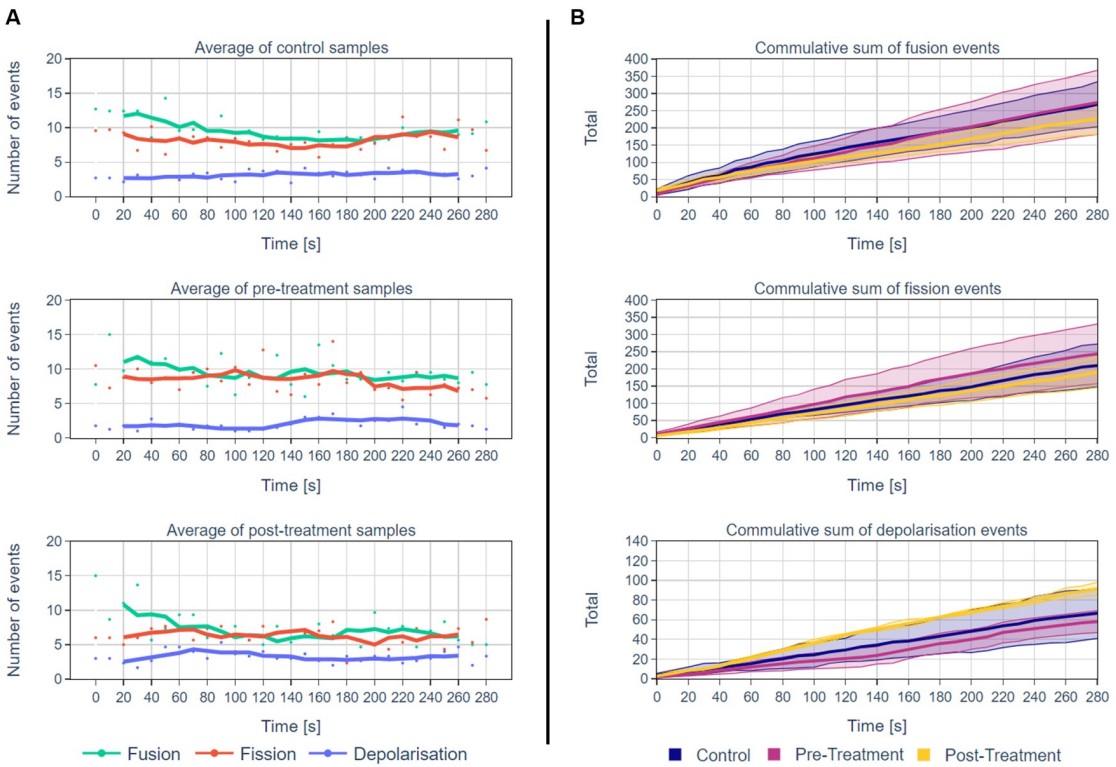

**Fig 10. A summary of MEL outputs for multiple independent single-cell analysis.** A: The average number of mitochondrial events for each time interval for the control cells and the cells before and after $H_2O_2$ treatment (n = 4 independent, biological repeats). The lines represent the five-frame simple moving average. B: A comparison of the cumulative sum of MEL detected mitochondrial events for the control, pre- and post-treatment groups. The mean event totals are thick lines and the 80% confidence intervals by shading (n = 4 independent, biological repeats). The fact that the confidence intervals overlap could be an indication that the sample size of four is too small. Note that the y-axis scales are different for the fusion/fission and depolarisation events.

groups. Fig 10B shows the mean of the cumulative sum of the MEL detected mitochondrial events for the different groups. Table 3 show mean, standard deviation for the number of mitochondrial fission, fusion and depolarisation events as well as the total number of mitochondrial structures with their average volume.

Next, we investigated whether statistically significant differences ($\alpha < 0.05$) could be detected between the different treatment groups when considering the MEL outputs of all cells

**Table 3. Comparison of the MEL outputs for multiple single-cell analysis under control conditions as well as before and after $H_2O_2$ treatment (n = 116 for each group).**

| | Control | | Pre-Treatment | | Post-Treatment | |
|---|---|---|---|---|---|---|
| Event Type | Mean | SD | Mean | SD | Mean | SD |
| Fusion | 9.3 | 5.5 | 9.4 | 5.2 | 7.8 | 4.4 |
| Fission | 7.2 | 4.2 | 7.8 | 4.4 | 6.6 | 3.0 |
| Depolarisation | 2.3 | 2.0 | 2.0 | 1.5 | 3.2 | 1.6 |
| Total Structures | 71.5 | 30.6 | 74.9 | 22.2 | 70.3 | 17.6 |
| Average Structure Volume | 1039.64 | 710.0 | 917.5 | 380.1 | 895.5 | 406.4 |

The means and standard deviations were calculated for all the cells in the different groups as separate measurements.

**Table 4. A comparison of the p-values produced by the two-sample, two-tailed t-test for the data summarised in Table 3.**

| Event type | Control vs. Pre-Treatment | Pre-Treatment vs. Post-Treatment | Control vs. Post-Treatment |
|---|---|---|---|
| Fusion | 0.800 | **0.005** | **0.030** |
| Fission | 0.058 | **p < 0.001** | 0.16 |
| Depolarisation | 0.210 | **p < 0.001** | **p < 0.001** |
| Total | 0.347 | 0.121 | 0.713 |
| Average | 0.104 | 0.706 | 0.059 |

The unpaired t-test was used to compare the control group with the pre-treatment and post-treatment groups. The paired t-test was used to compare the pre-treatment with the post-treatment groups. Statistically significant differences are highlighted in bold.

in the different groups as independent events. First, we used Bartlett's test for equal variances to select the appropriate two-sample, two-tailed t-test [27]. We then used the unpaired t-test to compare the control group with both pre- and post-treatment groups and the paired t-test to compare the groups before treatment with the group after treatment since they concerned the same cells. Table 4 summarises the p-values that were obtained from these test.

The results indicate that statistically significantly fewer fission and fusion events were detected in the post-treatment group compared to the pre-treatment group. This significant difference was also detected when comparing the fusion events for the control and post-treatment groups, however, for fission events we failed to reject the null hypothesis. Furthermore, statistically significantly more depolarisation were events detected in the post-treatment group than in the pre-treatment or control group.

## Discussion

We have described a novel method, MEL, which localises the key mitochondrial fission, fusion and depolarisation events in the three-dimensional space based on a time-lapse image sequence. We provide evidence that, when using MEL, the relationship between fission and fusion can be described quantitatively. This allows the assessment of whether a cellular system is in mitochondrial fission/fusion equilibrium and whether mitochondrial depolarisation events are kept within a cell specific physiological range. We also visualise where and when mitochondrial events, specifically fission, fusion and depolarisation, take place in the cell. Given the regional complexity of cells, which may distinctively change in disease, with a particular distribution profile of organelles, mitochondria, ATP generation and vesicle dynamics associated with mitochondrial quality control, MEL may be used to dissect localised regions of mitochondrial dysfunction or molecular defect precisely [28–32]. Moreover, when applied in scenarios where particularly mitochondrial pathology is implicated, such as in neurodegeneration or mitophagy, precise mitochondrial event localisation as provided by MEL may be of benefit [33]. Finally, due to the relatively simple data generation approach, a role for diagnostics, especially in the context of high-throughput imaging approaches, may be envisaged [34].

The current alternative to MEL involves the manual inspection of time-lapse images sequence and discovery of events by eye. This is not only very time consuming, but is also likely to miss some mitochondrial events, due to the complexity of the mitochondrial network and their wide distribution across the cell. MEL allows the automatic identification and subsequent visualisation of mitochondrial events over the time course of sample acquisition. This enables the dynamics of such mitochondrial activity to be captured automatically for subsequent quantitative analysis. False-positives, can be removed by manually evaluating the detected event using the validation tool. This increases the accuracy, especially for single-cell

analysis. Furthermore, as illustrated in Fig 10 and in Tables 3 and 4, by applying MEL to several samples in different treatment groups, these groups can be compared to discover differences.

When validating the MEL events, true-positive fission and fusion events varied between 38-58% (Table 2). The accuracy of the fission and fusion event detection is typically similar, which is likely a result of the back-and-forth structure matching algorithm (Fig 3) being the same for both types of events. True-positive depolarisation events were lower with percentages between 6-36% (Table 2). This relatively low percentage is due the high motility of small mitochondrial structures which prevent MEL from matching these structures between two consecutive time-lapse frames (see Fig 5). In some cases, small mitochondrial structures may also lack sufficient signal intensity in one of the two frames to be binarised as a mitochondrial structure also leading to a false-positive depolarisation event.

From a morphological point of view, we observe a widely spread, highly interconnected mitochondrial network for the control and pre-treatment samples (column A in Figs 7 and 8). The $H_2O_2$-treated samples (Figs 9 and 10) shows a distinct profile for specifically the fusion events. An initial bias toward fusion events (green) was observed as the network morphology becomes more densely networked, indicating a mitochondrial stress response or $H_2O_2$-induced mitochondrial injury. This results in fewer mitochondrial structures, each with a higher average volume. Unlike the control and pre-treatment samples, the change in the total number of mitochondrial structures as well as their average volume seems to slow after approximately 90 seconds. At this point, the cell appears to have reached a new fission/fusion equilibrium and remains stable until the end of our time-lapse sequence (at 290 seconds).

In general, there is very little change in the overall mitochondrial network pattern of the control and pre-treatment samples (column A in Figs 7 and 8). In sharp contrast, the mitochondrial network of the $H_2O_2$-treated samples (column A in Fig 9) initially appears more loosely networked and, as time progresses, is seen to become denser and more centred. This accounts for the decrease in the mitochondrial events. This is also confirmed the lower cumulative number of fission and fusion events as shown in Fig 10B.

## Fission and fusion events

Based on the events that were detected by MEL, both before and after false-positives were removed, as shown in column D in Figs 7–9 and Figs 6 and 10, we conclude that both the control and treated cells are characterised by the maintenance of a fission/fusion equilibrium throughout, with a slight bias toward fusion events.

From Tables 3 and 4 the number of fission and fusion events are statistically significantly fewer for the post-treatment group compared to both the control and pre-treatment group. This is primarily attributed to the cells becoming more densely networked as a stress response to the $H_2O_2$ treatment.

Qualitatively, most of the fission and fusion events were observed in the areas of the mitochondrial network which are more loosely networked, with very few events detected in the strongly networked areas. Previous work has indeed suggested that the density of the mitochondrial network influences fission and fusion events [20].

Using the control cell shown in Fig 7 as an example, MEL can also aid in the detection of mitochondrial structures that alternate between fission and fusion events as shown in Fig 11. This scenario is supported by previous work [35, 36], and is also referred to as 'kiss-and-run', albeit without the 3D quantitative approach to distinguish between structures that fuse and structures that move past each other at different depths.

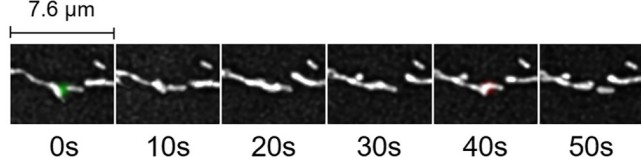

**Fig 11. An example of MEL detecting kiss-and-run in the control cell showcased in Fig 7.**

## Depolarisation events

To our knowledge, MEL is the first approach that allows the automatic localisation of depolarisation events in the three-dimensional context, especially within the spatio-temporal context of fission and fusion events. This is specifically significant since depolarisation of the entire mitochondrial network has been associated with caspase-3 activation, the execution of apoptotic cell death [37] and generally demarcates the point of no return (PONR) for apoptosis [38]. Therefore, MEL may be of particular value in the quantitative assessment of cell death onset, associated with a wide range of diseases.

From Tables 3 and 4, only a limited number of depolarisation events were detected throughout the time-lapse sequence. However, depolarisation was observed to be higher to a statistically significantly degree after $H_2O_2$ treatment than both before treatment for the same cell as well as compared to the control cells. This can also be seen in the cumulative sum of depolarisation events shown in Fig 10B.

Qualitatively, most depolarisation events were localised in mitochondria that become isolated from the main mitochondrial network and have a small average volume.

Small mitochondrial structures, however, pose a challenge MEL since they are difficult to track accurately from one frame to the next, especially if exhibit high motility. This has adverse effects on the accuracy with which depolarisation is detected and might explain the lack of statistically significant differences between control and post-treated cells.

## Future work

Future work may employ MEL when utilising additional mitochondrial probes, so as to include additional functional parameters in context of fission and fusion [39]. Cell variability can further be decreased using methods such as cell cycle synchronisation [40] or micropatterning approaches [41]. Future studies can also use MEL to investigate the potential of subcellular localisation of mitochondrial events.

## Conclusion

Mitochondrial events, particularly fission, fusion and depolarisation, play a central role in cellular homeostasis, function and viability. However, the precise localisation of these events within a cell as well as the quantitative characterisation thereof remains challenging.

Building upon existing tools for quantitative mitochondrial analysis [20], we have presented MEL (mitochondrial event localiser), an approach developed to automatically determine the three-dimensional location of fission, fusion and depolarisation events in a sample from a three-dimensional time-lapse sequence. The number of mitochondrial structures, their combined and average volume, as well as the number of events in each frame of the time-lapse sequence can subsequently be calculated to provide a quantitative assessment. The

mitochondrial events can also be visualised by superimposing them onto the fluorescence micrograph z-stack at the detected location.

To demonstrate its performance, we have applied MEL to three-dimensional time-lapse sequences of control cells as well as cells before and after hydrogen peroxide treatment. The analysis confirmed that the fission/fusion equilibrium is maintained throughout the time-lapse of the control and pre-treatment cells, as is commonly expected for healthy cells. The treated cells, on the other hand, showed an initial response toward fusion, causing the mitochondrial network to become denser and more localised in the perinuclear region.

Due to inaccuracies in the binarisation step, MEL can produce false-positive event detections. These false-positive detections can, however, be manually removed by using an accompanying tool that allows each detected event to be validated individually. We hope future work would build upon MEL to enhance its accuracy. Furthermore, when conducted using fully automated high throughput imaging platforms with precision liquid handling capabilities [42], further enhancement of MEL can be anticipated.

We conclude that MEL is a viable method of quantitative mitochondrial event analysis. This may consequently aid diagnosis and treatment strategies to be implemented for human pathologies associated with mitochondrial dysfunction.

## Supporting information

**S1 Fig. An example of a typical intensity histogram for the cells analysed with the low and high hysteresis threshold values indicated.** The low threshold is automatically calculated at the edge of the histogram valley of the background voxel intensities, and the high threshold at the halfway point between the low threshold and the maximum intensity.
(PNG)

**S2 Fig. An illustration of the limited effect of photo bleaching.** Control cells were imaged every 10 seconds for a total of 10 minutes. In order to illustrate the limited effect of photo bleaching, control cells that were acquired after 0 sec, 290 sec, and 590 sec are compared.
(TIF)

**S1 Appendix. Application of MEL to a synthetic example.**
(PDF)

**S2 Appendix. General considerations when using MEL.**
(PDF)

**S1 File.**
(ZIP)

## Acknowledgments

The authors wish to thank the Cell Imaging Unit, Central Analytical Facility (CAF), Stellenbosch University for providing technical support.

## Author Contributions

**Conceptualization:** Rensu P. Theart, Ben Loos, Thomas R. Niesler.

**Data curation:** Rensu P. Theart, Jurgen Kriel, André du Toit, Ben Loos.

**Formal analysis:** Rensu P. Theart.

**Funding acquisition:** Ben Loos, Thomas R. Niesler.

**Investigation:** Rensu P. Theart.

**Methodology:** Rensu P. Theart, Ben Loos.

**Project administration:** Ben Loos, Thomas R. Niesler.

**Resources:** Ben Loos, Thomas R. Niesler.

**Software:** Rensu P. Theart.

**Supervision:** Ben Loos, Thomas R. Niesler.

**Validation:** Rensu P. Theart, Ben Loos.

**Visualization:** Rensu P. Theart.

**Writing – original draft:** Rensu P. Theart, Jurgen Kriel, Ben Loos, Thomas R. Niesler.

**Writing – review & editing:** Rensu P. Theart, Ben Loos, Thomas R. Niesler.

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
