## [Decision Letter · Decision Letter 0]

1 Apr 2020

PONE-D-20-03864

Mitochondrial event localiser (MEL) to quantitatively describe fission, fusion and depolarisation in the three-dimensional space

PLOS ONE

Dear Mr Theart,

Thank you for submitting your manuscript to PLOS ONE. After careful consideration, we feel that it has merit but does not fully meet PLOS ONE’s publication criteria as it currently stands. Therefore, we invite you to submit a revised version of the manuscript that addresses the points raised during the review process. The study is of potential interest and I think it could provide a valuable tool.

Still, there are some major concerns that need to be addressed sufficuently. In particular, you need to provide a better quantitative validation of the analysis including the issue raised about possible false positives. Moreover, the PLOS One rules concerning availabily of algorithms, also raised by one reviewer, has to be addressed.

We would appreciate receiving your revised manuscript by May 16 2020 11:59PM. To enhance the reproducibility of your results, we recommend that if applicable you deposit your laboratory protocols in protocols.io, where a protocol can be assigned its own identifier (DOI) such that it can be cited independently in the future. For instructions see: http://journals.plos.org/plosone/s/submission-guidelines#loc-laboratory-protocols

We look forward to receiving your revised manuscript.

Kind regards,

Andreas S. Reichert, Dr. rer. nat.

Academic Editor

PLOS ONE

Journal Requirements:

2. Thank you for stating the following in the Competing Interests/Financial Disclosure * (delete as necessary) section:

"This work was supported by the South African Medical Research Council (SAMRC), Cancer Association of South Africa (CANSA), and Telkom South Africa, and the South African National Research Foundation (NRF). Nvidia corporation sponsored the GPU that was used in this project. The funders had no role in study design, data collection and analysis, decision to publish, or preparation of the manuscript."

We note that you received funding from a commercial source: Nvidia corporation

Reviewers' comments:

Reviewer's Responses to Questions

**Comments to the Author**

1. Is the manuscript technically sound, and do the data support the conclusions?

Reviewer #1: Yes

Reviewer #2: Partly

2. Has the statistical analysis been performed appropriately and rigorously? 

Reviewer #1: No

Reviewer #2: No

3. Have the authors made all data underlying the findings in their manuscript fully available?

Reviewer #1: No

Reviewer #2: Yes

4. Is the manuscript presented in an intelligible fashion and written in standard English?

Reviewer #1: Yes

Reviewer #2: Yes

5. Review Comments to the Author

Reviewer #1: The manuscript entitled "Mitochondrial event localiser (MEL) ..." by Theart et al. describes an automated analysis tool for 3D image stacks of mitochondrial fusion, fission and depolarisation. The manuscript is very well written and pleasent to read. The scientific topic of this manuscript is highly important as - to the best of my knowledge - an objective and automated analysis tool does not exist today; yet it is hihgly desirable, because the mitochondrial dynamics can only be accurately quantified in this way. The quantification of these processes is highly important, because it provides a firm basis for models of mitochondrial dynamics.

I have a number of serious issues that need to be addressed before the manuscript may be considered for publication:

1) In line 166-169, the authors list the parameters that enter the image analysis. I suggest that the authors give here the range in which they have tested these parameters for each of them. Absolute numbers are required in order to understand statements that are otherwise meaningless, because one does not know what they compare to; e.g.

"a time interaval that is too long" (line 112) - what is long and what not?;

"less than some appropriately small number of voxels (in our case 20)" - what is appropriate? how does one decide about this? what does it mean in absolute terms, i.e. in µm?

"overlapping volumes that account for less than 1% of the volume ... are eliminated" (caption of Fig. 2) - what makes 1% the magic number?

"with a small percentage" (line 228) - what is small? what is large? how to decide this?

2) Line 305-306: I do not understand why the experiments and controls were recorded with different time steps. One would expect that every condition is recorded with identical settings and it would be the best if time steps would be chosen as small as possible. This would not only increase accuracy of the analysis, but would also allow to systematically study the impact of increasing time steps: for experiment and control the analysis could be repeated with larger time steps (leaving frames in between out) - but still having the exact time stepping in experiment and control. Otherwise, I do not see how to uniquely attribute changes in the analysis performance to the difference in the experiment versus control, because such differences may as well stem from technical issues like different time stepping in the imaging. While having such differences should be avoided in the experimental design, the reasoning of the authors for their data set with incommensurate time steps is not at all clear.

3) In line 348ff, the visualization in virtual reality is emphasized. I do not understand this point, may be; however, what I would have liked to see is a number of videos (e.g. with changing perspectives) that gives some idea how the dynamics looks in three spatial dimensions. Why did the author not opt for this possibility?

4) Related to the last issue, in general, I am missing a quantitative validation of the analyis tool. The Results section and the Discussion section contain a lot of statements that are not quantitatively supported by a rigorous validation. No comparison is made; neither to manually annotated data, nor by using another analysis tool (which may not have been developed for mitochondria, but for other data, e.g. for cell migration and interaction). Without any quantitative validation of this analysis tool, I cannot see how the value of the presented tool can be objectively judged. Without such an analysis, I think, this work can not be published. I would like to know, how many false positives and false negatives you have in detecting and not detecting events of fusion/fission/depolarisation; please, provide the numbers of your validation in terms of standard performance measures.

5) Once issue 4) has been done, the authors can remove statements like "it was often observed" (line 408), "From the analysis it seems that MEL is robust..." (line 426), "the false detection of these events is reduced" (line 432), "the best 3D quantification" (line 442), "it produces the most consistent results" (line 446) and replace them by quantiatively concrete statements. Please, check the whole manuscript for many of these kinds of statements, which all need to be removed and replaced by quantitative statements with clear reference as to how properties like "robust" and "best" are measured.

6) Table 2 contains a list of all tunable parameters, which need to be given in realistic units; the symbol \\sigma should not be used repeatedly for different quantitites.

7) The authors are not sufficiently motivating the need of their analysis tool: it is, in fact, mandatory to have such a tool in order to model the mitochondrial dynamics. Models of mitochondrial dynamics should be cited, e.g. a quick search revealed:

https://doi.org/10.1371/journal.pcbi.1002576

https://doi.org/10.1002/bies.201200125

https://doi.org/10.1371/journal.pcbi.1003108

https://doi.org/10.1371/journal.pone.0168198

...

All these models (and may be others as well) will benefit from a tool like you have developed here, underlining the importance of it once it has been appropriately validated.

8) I somehow miss the information where the code can be downloaded and where the data can be downloaded. Did I overlook this?

Reviewer #2: Overall, I think this is an interesting paper that with more rigorous analysis and experimental replicates with statistical analysis has the chance to contribute to the field of mitochondria morphometry.

My most concerning critique is that the authors present the data of a single biological experimental replicate per condition. How can the community be assured of the repeatability of your method? This must be addressed prior to publication. Please describe and discuss the variance between accuracy (including false positive and false negative events as discussed in the critique below), results, and biological implications.

—-How do you distinguish between events that are truly depolarization and loss of florescence signal from photobleaching? One way to describe this would be an overall signal to background fluorescence intensity from frame to frame, but discussion of the lower limit of detection of the algorithm is important given the assumption that a disappearing label is a depolarization event. An alternative approach could be the use of a covalently or otherwise permanently bound fluorophore (e.g MitoTracker Deep Red or fluorescent-tagged membrane bound protein).

—- I would argue that the tunable parameter of minimum voxel volume or the otsu thresholding sensitivity should be re-tuned. In both the regions of interest image series displayed in figures 4 and 5, multiple incorrect categorizations of events can be observed by looking at the entire time lapse. For instance, in figure 5, a more dimly lit small spherical shaped mitochondrion is identified by the algorithm to undergo depolarization 3 times but by the naked eye it’s appearance changes very little and yet it exists across the entire image series. Similarly, the fission and fusion events the authors describe as kiss and run do not convincingly separate and join. If this is representative of the algorithm’s performance, then the reader must conclude that the false positive event rate is far too high to be acceptable.

— To address the concerns just raised, the authors should (ideally under 3D visualization) manually determine the true positive, false positive and false negative rates of event localization and report these adjacent to the event rate graphs and discuss how this was used to tune their algorithm’s parameters.

— One of the tenets of the scientific method is to alter only one variable during an experiment. By increasing the temporal density of imaging in the control versus toxicant treatment group, the authors have introduced differing levels of photobleaching and phototoxicity as a variable as well as varied the potential for missing events or mismatching labeled structures. Please address this time difference by repeating your experiments under similar conditions, address this bias in your discussion, or leave out any claims interpreting the changes observed as an effect of the toxicant.

— What was the vehicle? We have observed empirically that mitochondria are exquisitely sensitive to dimethyl sulfoxide in any final concentration greater than 0.1%. Please list the vehicle and its concentration in your methods and ensure it is also administered at the same concentration in your control population of cells at an identical time prior to imaging.

6. PLOS authors have the option to publish the peer review history of their article (what does this mean?). If published, this will include your full peer review and any attached files.

Reviewer #1: No

Reviewer #2: No

---

## [Author Response · Author response to Decision Letter 0]

1 Oct 2020

Responses to the reviewer's comments for the submission:

“Mitochondrial event localiser (MEL) to quantitatively describe fission, fusion and depolarisation in the three-dimensional space”.

PONE-D-20-03864

Theart, Kriel, du Toit, Loos, and Niesler 

University of Stellenbosch

South Africa

We would like to thank the reviewers for their effort in reviewing our manuscript. The comments have been highly constructive, allowing us to consider the details of the method we present more closely and state our design choices more clearly. We have carefully addressed the recommendations. 

Response to the editor

In response to the points raised by the editor, in the revised manuscript:

• We have provided quantitative validation of MEL and summarised these results in Table 1 and Fig 6. We have also indicated and described common scenarios in which false-positive events are detected (Fig 5). Furthermore, we developed a validation tool with which false positive events can be removed prior to generating the final output number of mitochondrial events and locations. This tool is included in the code submitted for the revision.

• We have now provided the full source code that implements the algorithm in the revision. The entire manuscript has been carefully revised and edited. 

In the following text, reviewers' verbatim comments are shown first, and our responses follow.

Reviewer 1.

Reviewer's comment 1.

The manuscript entitled "Mitochondrial event localiser (MEL) ..." by Theart et al. describes an automated analysis tool for 3D image stacks of mitochondrial fusion, fission and depolarisation. The manuscript is very well written and pleasent to read. The scientific topic of this manuscript is highly important as - to the best of my knowledge - an objective and automated analysis tool does not exist today; yet it is highly desirable, because the mitochondrial dynamics can only be accurately quantified in this way. The quantification of these processes is highly important, because it provides a firm basis for models of mitochondrial dynamics.

Authors' response 1.

We thank the reviewer for the kind words and also acknowledging the importance of the problem that we set out to solve in this paper. We are also not aware of an automated analysis tool that can achieve this. The software implementation of our method has been included with the revision for use by other researchers.

Reviewer's comment 2.

I have a number of serious issues that need to be addressed before the manuscript may be considered for publication:

1) In line 166-169, the authors list the parameters that enter the image analysis. I suggest that the authors give here the range in which they have tested these parameters for each of them. Absolute numbers are required in order to understand statements that are otherwise meaningless, because one does not know what they compare to; e.g.

"a time interval that is too long" (line 112) - what is long and what not?;

"less than some appropriately small number of voxels (in our case 20)" - what is appropriate? how does one decide about this? what does it mean in absolute terms, i.e. in µm?

"overlapping volumes that account for less than 1% of the volume ... are eliminated" (caption of Fig. 2) - what makes 1% the magic number?

"with a small percentage" (line 228) - what is small? what is large? how to decide this?

Authors' response 2.

We appreciate the reviewer mentioning the lack of specificity of some assumptions made with MEL. We have now corrected these aspects, focussing on absolute numbers, and clearly indicating the statements as follows:

• Line 166-169: The three tunable parameters in question are the two different standard deviations used for the 2D and 3D Gaussian blurs and the volume of structures considered as noise. The values of these parameters were chosen empirically by considering their effect on the sample data that we analysed in this paper, and the ranges we provide might not be appropriate to all conditions and acquisition parameters. Regardless, we have adjusted the paragraph in question by adding the following text: “The volume should be determined empirically by considering the average size of the binarised noise without erroneously removing true mitochondrial structures. From our testing, the value of σ2D in the normalisation step should take values between 0.5 and 1.5. σ3D in the 3D Gaussian filter step should take values between 0.2 and 0.5 (0.02-0.04 µm in x-y and 0.1-0.25 µm in z) due to the limited resolution in the z-dimension. These values are, however, subject to different image acquisition parameters.”

• Line 112: We added the following sentence as explanation: “The required time interval for accurate results is dependent on the motility of the mitochondria, but in our experimentation time intervals of 10-30 seconds produced good results. Shorter time intervals such as 5 seconds may also be used.”

• What is appropriate?: We updated the section in which this comment was made as follows: “Since it has been observed that some noise remnants were also being binarised, we removed any structure containing less than 40 voxels in the upscaled images. We determined this value empirically by considering the average size of the binarised noise structures. In concrete terms, this is equivalent to a circle with a diameter of 5 pixels in the original image, or in our case a circular structure with a physical diameter of about 0.6 µm.”

• Fig 2 Caption and Line 228: We thank the reviewer for allowing us to reconsider our method design. In the process of refining the MEL method for the revised submission we realised that there was no need for the small percentage to remove ‘coincidental’ events due to slight overlaps between structures since the overlap percentages that are used at a later stage already eliminated these events. As this in no longer part of the method we have removed the corresponding sentences from the revised manuscript.

Reviewer's comment 3.

2) Line 305-306: I do not understand why the experiments and controls were recorded with different time steps. One would expect that every condition is recorded with identical settings and it would be the best if time steps would be chosen as small as possible. This would not only increase accuracy of the analysis, but would also allow to systematically study the impact of increasing time steps: for experiment and control the analysis could be repeated with larger time steps (leaving frames in between out) - but still having the exact time stepping in experiment and control. Otherwise, I do not see how to uniquely attribute changes in the analysis performance to the difference in the experiment versus control, because such differences may as well stem from technical issues like different time stepping in the imaging. While having such differences should be avoided in the experimental design, the reasoning of the authors for their data set with incommensurate time steps is not at all clear.

Authors' response 3.

We wish to thank the reviewer for this important feedback. These differences were due to distinct acquisition runs that were performed during the development of MEL, and we wished to showcase the application of MEL under various conditions.

We have now acquired substantial additional data sets, repeated imaging acquisitions under control and treatment conditions, keeping a consistent time interval of 10 seconds for both control and treated samples. Moreover, we have used the same cells pre- and post-treatment, which was technically not trivial, using a 100x oil immersion objective. Hence, this now enables a direct comparison of the different mitochondrial events under the conditions assessed. In addition to the control cells, the pre-and post-treated cells were also compared which allowed the assessment of dynamic changes of the mitochondrial events, as opposed to merely endpoints. In order to induce a mitochondrial response, we have now utilized H2O2 , to achieve a generic and more commonly implemented cellular stress response. Care was taken to implement similar and controlled acquisition parameters that favour dynamic range, minimize bleaching and allow rapid acquisition in z with high spatial and temporal resolution.

Reviewer's comment 4.

3) In line 348ff, the visualization in virtual reality is emphasized. I do not understand this point, may be; however, what I would have liked to see is a number of videos (e.g. with changing perspectives) that gives some idea how the dynamics looks in three spatial dimensions. Why did the author not opt for this possibility?

Authors' response 4.

We thank the reviewer for the comment. The MEL method is independent of the use of virtual reality, and in the initial submission we resorted to using a previously developed virtual reality tool that enables precise three-dimensional region of interest selections. 

As to not distract from the focus of the manuscript, we have now removed the reference to virtual reality and instead performed the 3D ROI selections and visualisation with Fiji (ImageJ).

Reviewer's comment 5.

4) Related to the last issue, in general, I am missing a quantitative validation of the analysis tool. The Results section and the Discussion section contain a lot of statements that are not quantitatively supported by a rigorous validation. No comparison is made; neither to manually annotated data, nor by using another analysis tool (which may not have been developed for mitochondria, but for other data, e.g. for cell migration and interaction). Without any quantitative validation of this analysis tool, I cannot see how the value of the presented tool can be objectively judged. Without such an analysis, I think, this work can not be published. I would like to know, how many false positives and false negatives you have in detecting and not detecting events of fusion/fission/depolarisation; please, provide the numbers of your validation in terms of standard performance measures.

Authors' response 5.

We thank the reviewer for this important aspect. We have now performed an extensive analysis on the large, newly acquired data sets, so as to better indicate quantitative validation of the analysis tool. As far as has been possible, we have assessed the number of false-positive events. 

The type of analysis that is performed by MEL is, however, challenging to perform accurately ‘by eye’, especially on complex networked mitochondrial structures in three-dimensional space. Although we were not able to locate a tool that achieves something similar enough to be used as a comparison with MEL, we have developed a tool, which has been submitted as part of the revision, that life science experts used to verify the mitochondrial events detected by MEL individually. This allowed us to determine the number of false positives that were detected by MEL and it is hoped to address the reviewer’s quantitative validation concern.

Due to the incredible abundance of such events for multiple cells over a time-lapse sequence we have limited this validation to a data set of a representative cell from the control, pre- and post-treated groups. The quantitative validation results are summarised in Table 1 and Fig 6. In addition, to enhance clarity and user implementation, we have now indicated and described common scenarios in which false-positive events are detected (Fig 5). Our analysis confirms that MEL operates favourably, especially in the context of detecting changes in mitochondrial events over time and comparing different treatment groups with each other. In addition, we have now included a supplementary section titled “General considerations for MEL” which outlines technical considerations that may assist the user in generating raw data that are least prone to inaccurate analysis, for example, due to poor thresholding. This may serve as a transparent point of departure for further implementation by the imaging community.

Moreover, we have added the comment of validating the events to line 105 in the method:

“Since some of the detected events could be false positives as a consequence of the thresholding step joining two mitochondrial structures together that are close to each other but not in fact fused, MEL also allows for each event to be subsequently validated and removed from the visualisation and event counts.” 

Reviewer's comment 6.

5) Once issue 4) has been done, the authors can remove statements like "it was often observed" (line 408), "From the analysis it seems that MEL is robust..." (line 426), "the false detection of these events is reduced" (line 432), "the best 3D quantification" (line 442), "it produces the most consistent results" (line 446) and replace them by quantiatively concrete statements. Please, check the whole manuscript for many of these kinds of statements, which all need to be removed and replaced by quantitative statements with clear reference as to how properties like "robust" and "best" are measured.

Authors' response 6.

We appreciate the reviewer’s comment. We have edited the comments in the manuscript as follows:

• Line 408: This statement has been removed and the text now reads: “MEL can also aid in the detection of mitochondrial structures that alternate between fission and fusion events as shown in Fig 11.”

• Line 426: This line has been removed since the statement is no longer relevant to the new data that we showcased.

• Line 432: We have changed the manuscript to read: “However, since MEL considers the entire three-dimensional structure of mitochondria, it can distinguish mitochondrial structures passing each other from true fusion between these structures. For this to be possible, the microscope that is used should have sufficient resolving power and the z-stack should contain enough micrographs. We found using a 100x oil immersion objective with the microscope and acquiring image stacks with 4-6 micrographs, with a 0.5 µm step width, produced favourable results.”

• Line 442: This statement was in reference to other work that was cited. We have, however, changed the text to read: “Such a strategy was used, for example, for an automated image analysis algorithm that determines the optimal filtering parameters to produce an optimised 3D quantification of mitochondrial morphology.”

• Line 446: This statement now reads: “It is for this reason that we first normalise the images before applying hysteresis thresholding to ensure it produces more consistent results.”

• We have also corrected similar statements throughout the manuscript.

Reviewer's comment 7.

6) Table 2 contains a list of all tunable parameters, which need to be given in realistic units; the symbol \\sigma should not be used repeatedly for different quantitites.

Authors' response 7.

We thank the reviewer for the comment and for picking up that we erroneously used \\sigma to represent different quantities. We have now distinguished between the two-dimensional Gaussian filter used during normalisation by using \\sigma_{2D} and the three-dimensional Gaussian filter as \\sigma_{3D}.

In the revised manuscript we have also included all the values in realistic units throughout the paper, as well as in S1 Table.

Reviewer's comment 8.

7) The authors are not sufficiently motivating the need of their analysis tool: it is, in fact, mandatory to have such a tool in order to model the mitochondrial dynamics. Models of mitochondrial dynamics should be cited, e.g. a quick search revealed:

https://doi.org/10.1371/journal.pcbi.1002576

https://doi.org/10.1002/bies.201200125

https://doi.org/10.1371/journal.pcbi.1003108

https://doi.org/10.1371/journal.pone.0168198

All these models (and may be others as well) will benefit from a tool like you have developed here, underlining the importance of it once it has been appropriately validated.

Authors' response 8.

We wish to thank the reviewer for this comment and the recommendations of referring to existing tools. 

We have now included these in the manuscript (line 24), so as to contextualize the strength and potential importance of MEL better. 

“Furthermore, major interest exists in quantitatively describing mitochondrial dynamics to better discern the role of mitochondrial dysfunction in pathology and pathogenesis of major human diseases and to unravel whether changes in fission and fusion are adaptive or maladaptive. This includes the role of mitochondrial function in ageing (Ref 1 and 2), in context of mitochondrial quality control and mitophagy (Ref 3) as well as in the context of mitochondrial directionality and network distribution (Ref 4). However, a tool that assists in the rapid characterization, localization and quantification of mitochondrial fission, fusion and depolarisation events in the subcellular, three-dimensional space is currently not available.”

The purpose of MEL has been clarified in the introduction in Line 77:

“[MEL] provides insight into the spatio-temporal change of mitochondrial events in a time-lapse sequence and can also be used to compare mitochondrial event dynamics under different treatment conditions”

Reviewer's comment 9.

8) I somehow miss the information where the code can be downloaded and where the data can be downloaded. Did I overlook this?

Authors' response 9.

We thank the reviewer for requesting the code. This was an oversight on our part, and we have accordingly submitted it as part of the revised manuscript.

Reviewer 2.

Reviewer's comment 1.

Overall, I think this is an interesting paper that with more rigorous analysis and experimental replicates with statistical analysis has the chance to contribute to the field of mitochondria morphometry.

My most concerning critique is that the authors present the data of a single biological experimental replicate per condition. How can the community be assured of the repeatability of your method? This must be addressed prior to publication. Please describe and discuss the variance between accuracy (including false positive and false negative events as discussed in the critique below), results, and biological implications.

Authors' response 1.

We wish to thank the reviewer for this important comment. We have now attended to this aspect and have carefully acquired large additional data sets, so as to allow better and direct comparison and statistical analysis. 

It is challenging to precisely evaluate the accuracy of MEL, since to our knowledge no tool exists currently which achieves something similar. To address the reviewer’s comment, we have developed a tool which was used by life science experts to verify the mitochondrial events detected by MEL individually. In this way we were able to determine the number of false positives that were detected by MEL under different conditions. The result of this is summarised in Table 1 and Fig 6. We have also indicated in reference to Fig 5 the common scenarios in which false-positive events are detected. Our analysis confirms that MEL operates favourably, especially in the context of detecting changes in the number of mitochondrial events and comparing different treatment groups with each other.

A supplementary section titled “General considerations for MEL” have also been added, in which we outline some technical considerations to generate raw data that are least prone to inaccurate analysis, for example, due to poor thresholding. This may serve as a transparent point of departure for further implementation by the imaging community.

We have also added the comment of validating the events to line 105 in the method:

“Since some of the detected events could be false positives as a consequence of the thresholding step joining two mitochondrial structures together that are close to each other but not in fact fused, MEL also allows for each event to be subsequently validated and removed from the visualisation and event counts.” 

Reviewer's comment 2.

How do you distinguish between events that are truly depolarization and loss of florescence signal from photobleaching? One way to describe this would be an overall signal to background fluorescence intensity from frame to frame, but discussion of the lower limit of detection of the algorithm is important given the assumption that a disappearing label is a depolarization event. An alternative approach could be the use of a covalently or otherwise permanently bound fluorophore (e.g MitoTracker Deep Red or fluorescent-tagged membrane bound protein).

Authors' response 2.

The reviewer raises an important point. For our analysis that was performed and is presented in the revised manuscript, we have now acquired data in a highly controlled environment, using a very rapid scanning speed, very low (down to 0.2) laser power and have optimized dynamic range and photonmultipliertube gain, so as to avoid photobleaching. We have, for that purpose, implemented a time-lapse sequence that is resolved and extensive enough, but does not result in bleaching of signal. There was therefore no requirement to compensate for the detection of depolarisation in such conditions.

In the revised manuscript we now make use of hysteresis thresholding which can distinguish better between background and dim mitochondrial structures. The lower limit of detection is therefore determined by the threshold values that are used. We have added the following explanation in Line 139:

“Hysteresis thresholding is very effective in removing background voxels that have intensities similar to the mitochondrial object voxels. It uses two threshold values. Voxels that have an intensity above the high threshold are considered to belong to the object and voxels that have an intensity below the low threshold are considered to belong to the background. Voxels that have an intensity between the low and high thresholds are considered to belong to the object if they are connected to other object voxels. We automatically calculated the low threshold at the edge of the histogram valley of the background voxels intensities, and the high threshold at the halfway point between the low threshold and the maximum intensity.”

We agree, a permanently bound fluorphore could equally be used for this analysis. Although we also frequently use MitoTrackers, for the purpose of this study, we chose TMRE, due to its wide use and application in assessing mitochondrial dysfunction, and due its exceptionally good signal-to-noise ratio. However, MEL would equally operate on image data acquired using other fluorescent probes. We have now included this in the manuscript, to enhance applicability and user range. 

Reviewer's comment 3.

I would argue that the tunable parameter of minimum voxel volume or the otsu thresholding sensitivity should be re-tuned. In both the regions of interest image series displayed in figures 4 and 5, multiple incorrect categorizations of events can be observed by looking at the entire time lapse. For instance, in figure 5, a more dimly lit small spherical shaped mitochondrion is identified by the algorithm to undergo depolarization 3 times but by the naked eye it’s appearance changes very little and yet it exists across the entire image series. Similarly, the fission and fusion events the authors describe as kiss and run do not convincingly separate and join. If this is representative of the algorithm’s performance, then the reader must conclude that the false positive event rate is far too high to be acceptable.

Authors' response 3.

We thank the reviewer for drawing our attention to these false-positive events.

We have now included a section in the revised manuscript where we discuss the detection of false-positive events and illustrate some common scenarios in Fig 5. The false detection of depolarisation of small structures is particularly difficult to solve reliably. We have now included the following statement:

“Depolarisation can sometimes be falsely detected when small mitochondrial structures move so far between Frame 1 and Frame 2 that MEL no longer considers them to be associated.”

MEL is strongly dependent on the quality of the binarised image frames after applying a thresholding algorithm. Accurate thresholding is, however, a common challenge in microscopy and image analysis research and often poorly addressed and standardised. For the revised manuscript, we have however moved away from Otsu thresholding and have instead used Hysteresis thresholding which can distinguish better between background and mitochondrial structures. We have added the following explanation in Line 139:

“Hysteresis thresholding is very effective in removing background voxels that have intensities similar to the mitochondrial object voxels. It uses two threshold values. Voxels that have an intensity above the high threshold are considered to belong to the object and voxels that have an intensity below the low threshold are considered to belong to the background. Voxels that have an intensity between the low and high thresholds are considered to belong to the object if they are connected to other object voxels. We automatically calculated the low threshold at the edge of the histogram valley of the background voxels intensities, and the high threshold at the halfway point between the low threshold and the maximum intensity.”.”

Lastly, we have also observed a clearer example of the kiss-and-run phenomenon and have included it in the Discussion section as Fig 11. 

Reviewer's comment 4.

To address the concerns just raised, the authors should (ideally under 3D visualization) manually determine the true positive, false positive and false negative rates of event localization and report these adjacent to the event rate graphs and discuss how this was used to tune their algorithm’s parameters.

Authors' response 4.

We have now developed a validation tool with which each individual mitochondrial event can be validated manually. A screenshot of this tool is shown in the newly added Fig 4. The investigator can both cycle through the micrographs in the z-stack in the cropped region of the event to compare two consecutive time-lapse frames as well as consider the three-dimensionally reconstructed view of the structures expected to undergo the event. By using this tool, we were able to test various algorithm parameters to determine the optimal parameters for our dataset empirically.

Using this method, by considering each event individually, we noticed frequent false-positive fission and fusion events, primarily due to thresholding. To improve thresholding accuracy, effort has therefore been made to improve the normalisation and pre-processing of the input z-stacks, as well as changing the thresholding method from Otsu to Hysteresis thresholding.

Reviewer's comment 5.

One of the tenets of the scientific method is to alter only one variable during an experiment. By increasing the temporal density of imaging in the control versus toxicant treatment group, the authors have introduced differing levels of photobleaching and phototoxicity as a variable as well as varied the potential for missing events or mismatching labeled structures. Please address this time difference by repeating your experiments under similar conditions, address this bias in your discussion, or leave out any claims interpreting the changes observed as an effect of the toxicant.

Authors' response 5.

We wish to thank the reviewer for this important comment. We have now generated an extensive new data set, where care has been taken to implement absolutely similar acquisition parameters. In addition, we have now utilized very low laser power (down to 0.2) and a very rapid acquisition speed, acquiring only the cell in a particular field of view, so as to further decrease acquisition time. The image sequence and time interval has been adjusted, so that no photobleaching is introduced to the sample. A consistent time interval of 10 seconds for control, pre- and post-treated samples have been chosen. Moreover, we have used the same cells pre- and post-treatment, which was technically not trivial. Hence, this now enables a better and direct comparison of the different mitochondrial events under the conditions assessed. 

Reviewer's comment 6.

What was the vehicle? We have observed empirically that mitochondria are exquisitely sensitive to dimethyl sulfoxide in any final concentration greater than 0.1%. Please list the vehicle and its concentration in your methods and ensure it is also administered at the same concentration in your control population of cells at an identical time prior to imaging.

Authors' response 6.

We appreciate the reviewer's comment. Absolutely, we are also crucially aware of the incredible sensitivity of the mitochondrial network, including slightest changes in temperature, pH and solvents used. Here, TMRE has been utilized and made up in distilled water. 

We can also include a sentence in the method section, where we indicate the fluorescent probes, eg ‘Care should be taken to not perturb the cellular microenvironment, due to, for example, the solvent used, T or pH fluctuations, since the mitochondrial network will be impacted’. 

General paper edits.

In addition to the implemented changes as recommend by the reviewers, we have carefully assessed the entire manuscript for minor typographical, stylistic and other errors. These have been corrected. Additional references have been included, so as to better contextualize the advantages of MEL, and to better stress its potential relevance in the biomedical and life science field. Substantial effort has been made to implement MEL using a large, newly generated data set, with carefully controlled conditions. Additional comments and notes have been included in the manuscript, so as to enhance transparency, indicate potential pitfalls and to overall strengthen implementation of MEL in the imaging community. 

We have supplied both the revised manuscript as well as a version showing where these edits were undertaken in our resubmission.

---

## [Decision Letter · Decision Letter 1]

26 Oct 2020

PONE-D-20-03864R1

Mitochondrial event localiser (MEL) to quantitatively describe fission, fusion and depolarisation in the three-dimensional space

PLOS ONE

Dear Dr. Theart,

Thank you for submitting your manuscript to PLOS ONE. After careful consideration, we feel that it has merit but does not yet fully meet PLOS ONE’s publication criteria as it currently stands. Therefore, we invite you to submit a revised version of the manuscript that addresses the points raised during the review process.

I think that the study has improved very much and many of the most critical points are resolved. Still, I kindly ask you to adress the remaining concerns and submit a revised version. Could you please respond specifically in which case biological replicates were generated and if not why the method is still sufficiently validated.

We look forward to receiving your revised manuscript.

Kind regards,

Andreas S. Reichert, Dr. rer. nat.

Academic Editor

PLOS ONE

Reviewers' comments:

Reviewer's Responses to Questions

**Comments to the Author**

1. If the authors have adequately addressed your comments raised in a previous round of review and you feel that this manuscript is now acceptable for publication, you may indicate that here to bypass the “Comments to the Author” section, enter your conflict of interest statement in the “Confidential to Editor” section, and submit your "Accept" recommendation.

Reviewer #1: All comments have been addressed

Reviewer #2: (No Response)

2. Is the manuscript technically sound, and do the data support the conclusions?

Reviewer #1: Yes

Reviewer #2: Partly

3. Has the statistical analysis been performed appropriately and rigorously? 

Reviewer #1: Yes

Reviewer #2: No

4. Have the authors made all data underlying the findings in their manuscript fully available?

Reviewer #1: Yes

Reviewer #2: Yes

5. Is the manuscript presented in an intelligible fashion and written in standard English?

Reviewer #1: Yes

Reviewer #2: Yes

6. Review Comments to the Author

Reviewer #1: The authors have fully addressed my concerns, including the extension of the data set and the execution of a quantitative validation.

Reviewer #2: Major critiques:

The revision addresses most of my minor critiques but fails to address my primary concern that the authors have not proven the repeatability of the biological findings they are claiming with repeat experimental replicates. While they do describe their n in terms of number of cells analyzed, I cannot find anywhere in the paper that suggests the biological experiment was repeated more than once under identical conditions for each condition. This unfortunately remains a fatal flaw that must be addressed prior to publication. If this is what was done for Fig. 10, please make it abundantly clear to the reader that these were independent experiments.

Abstract:

The high rate of error should be addressed as a limitation up front so as not to surprise the reader in the results. The authors make the argument that the error of the algorithm in correctly detecting fission fusion and depolarization events in their peroxide-induced stress experiment correlates with the ground truth data as determined by manual review but with false positive rates ranging from at best 84% to 2000% (taken by dividing the means on the left side by the right side on Table 2), this is a huge limitation.

Please present the critical quantitative findings of the paper from Fig. 10 in the abstract or consider changing the title from quantitative to qualitative. E.g., “an average of x/x/x fusion/fission/depolarisation events per cell (verified/unverified?) were observed every 10 sec in control cells. With peroxide treatment, the rate initially rebalanced toward fusion to x/x/x, before returning to approximately the same as control levels x/x/x.”

See first comment below on discussion. Consider removing the subcellular localization comments from the abstract. They distract from the main points of the paper.

Results:

Why is data collected on control cells for only ~250 seconds when it appears that the treated group is imaged for twice as long (combining pre treatment and post treatment)? This opens up the opportunity for confounding variables including photo bleaching and photooxidative stress. Was vehicle added to the control cells at the same time point as the treated cells?

When it comes to statistical analysis, I’m not sure that lumping all of the many time-resolved events together via an average makes the most convincing argument for differences. While it is great and important to show the very high temporal resolution of the method, it would be more interesting and probably more statistically powerful to look at the summation of events over time (on a per cell basis) as opposed to averaging over time. Other alternatives would be analyzing specific time points or the derivative/rate of change.

——

Minor critiques:

Figure 6:

The control and post treatment time lapse curves appear identical. Please verify.

Secondly, the panels at the very bottom employ the use of a line graph that suggests a paired nature to the data. Bar graphs grouped by each event type would make the data easier to interpret to the reader. The error bars in the updated figure should be error between experimental replicates, not error between cells within an experimental replicate. Please add text to clarify that the number of events are per cell or per high powered field.

Why are the control events so different from the pre-treated cells? It makes the reader wonder whether there is a high amount of biological variability (which could be addressed with repeat experiments) or a confounding variable has not been addressed in the experimental set up (temperature, CO2, pH?).

Figure 7-9:

Why is there such a difference in the total number of detected structures in the last frame of untreated versus the first frame of the hydrogen peroxide-treated cells? The appearance of the cell has significantly changed suggesting that time 0 is not exactly the time at which peroxide was administered.

Furthermore, the overall cell morphology of the pre-treated and the control cell is completely different. I infer from the 2D representation that the pre-treated cell is more rounded and less adherent to the culture plate which is in contrast to the control which has the absence of mitochondria within the nuclear borders and spindle-shaped morphology and is adjacent to another cell. This suggests a different cell state which in and of itself can cause vastly different responses to treatment as well as baseline cell signaling characteristics. I would verify culture conditions are exactly the same between the groups. However, for representation of data, I would suggest picking cells that appear similar at baseline to show your reader.

Discussion:

I would refrain from making statements about patterns observed that aren’t rooted in multiple observations with quantification across biological replicates with statistical testing of your hypothesis. In particular, making claims about sub cellular localization of events in reference to peri nuclear versus periphery when no distance data is presented in the paper may make the reader question your findings. Instead, I would suggest alluding how your method could be used to test those hypotheses in future studies.

Conclusion:

Other methods (including those you referenced) have achieved deterministic automated qualification of mitochondrial morphology, albeit in in 2D not 3D. I would qualify your closing statements so as to not discount the work of others.

7. PLOS authors have the option to publish the peer review history of their article (what does this mean?). If published, this will include your full peer review and any attached files.

Reviewer #1: No

Reviewer #2: **Yes: **Anthony Presley Leonard

---

## [Author Response · Author response to Decision Letter 1]

10 Dec 2020

Responses to the reviewer's comments for the revised submission:

“Mitochondrial event localiser (MEL) to quantitatively describe fission, fusion and depolarisation in the three-dimensional space”.

PONE-D-20-03864

Theart, Kriel, du Toit, Loos, and Niesler 

University of Stellenbosch

South Africa

We would like to thank the reviewers for their effort in reviewing our paper. The comments have been highly constructive, allowing us to consider the details of the method we present more closely and state our design choices more clearly. We have carefully addressed the recommendations. 

Response to the editor

In response to the points raised by the editor, in the revised manuscript we have now clarified in line 419 as well as in the caption of Fig 10 that the results are indeed independent, biological repeats. For this reason, we believe that the method is sufficiently validated.

In the following text, reviewers' verbatim comments are shown first, and our responses follow.

Reviewer 2.

Major critiques:

Reviewer's comment 1.

The revision addresses most of my minor critiques but fails to address my primary concern that the authors have not proven the repeatability of the biological findings they are claiming with repeat experimental replicates. While they do describe their n in terms of number of cells analyzed, I cannot find anywhere in the paper that suggests the biological experiment was repeated more than once under identical conditions for each condition. This unfortunately remains a fatal flaw that must be addressed prior to publication. If this is what was done for Fig. 10, please make it abundantly clear to the reader that these were independent experiments.

Authors' response 1.

We wish to thank the reviewer for indicating this important aspect. We have now made it clear in line 419 as well as in the caption of Fig 10, that the results are indeed independent, biological repeats. For each biological repeat, newly passaged cells were seeded and subsequently imaged. Since we utilised 8-chamber dishes, with each independent experiment, also technical repeats were acquired. 

Moreover, we are now also referring to additional avenues (line 528) that may be considered in future, to further decrease variability, such as cell cycle synchronisation, or, as established in our laboratory, micropatterning.

Reviewer's comment 2.

Abstract:

The high rate of error should be addressed as a limitation up front so as not to surprise the reader in the results. The authors make the argument that the error of the algorithm in correctly detecting fission fusion and depolarization events in their peroxide-induced stress experiment correlates with the ground truth data as determined by manual review but with false positive rates ranging from at best 84% to 2000% (taken by dividing the means on the left side by the right side on Table 2), this is a huge limitation.

Authors' response 2.

We wish to thank the reviewer for this comment. We have now addressed this limitation in the abstract upfront by adding the following sentence: 

“When individually validating mitochondrial events detected with MEL, for a representative cell for the control and treated samples, the true-positive events were 47%/49%/14% respectively for fusion/fission/depolarisation events.”

In addition, we have also added a new table (Table 2) to indicate the accuracy for the different conditions, and enhanced clarity in the text (line 470). 

Reviewer's comment 3.

Please present the critical quantitative findings of the paper from Fig. 10 in the abstract or consider changing the title from quantitative to qualitative. E.g., “an average of x/x/x fusion/fission/depolarisation events per cell (verified/unverified?) were observed every 10 sec in control cells. With peroxide treatment, the rate initially rebalanced toward fusion to x/x/x, before returning to approximately the same as control levels x/x/x.”

Authors' response 3.

We thank the reviewer for this recommendation. We have now presented the critical quantitative findings in the abstract by adding the following text:

“An average of 9.3/7.2/2.3 fusion/fission/depolarisation events per cell were observed respectively for every 10 sec in the control cells. With peroxide treatment, the rate initially shifted toward fusion with an average of 15/6/3 events per cell, before returning to a new equilibrium not far from that of the control cells, with 6.2/6.4/3.4 events per cell.”

Reviewer's comment 4.

See first comment below on discussion. Consider removing the subcellular localization comments from the abstract. They distract from the main points of the paper.

Authors' response 4.

We thank the reviewer for this comment. We have now removed the subcellular localisation comments from the abstract, so as to enhance focus.

Reviewer's comment 5.

Results:

Why is data collected on control cells for only ~250 seconds when it appears that the treated group is imaged for twice as long (combining pre treatment and post treatment)? This opens up the opportunity for confounding variables including photo bleaching and photooxidative stress. Was vehicle added to the control cells at the same time point as the treated cells?

Authors' response 5.

We thank the reviewer for this important point. Indeed, all cells were acquired for a period of 300 seconds. A five-frame moving average was employed in the graphs in Figs 6-10 and is mentioned in the text in line 412.

We also recognise that the difference between the acquisition time between the control and treated cells were not clearly explained. We have now carefully described this in the manuscript (line 345):

“Low laser power was used to ensure that photo bleaching and photooxidative stress was limited during this acquisition period. Since the total acquisition time for the treated cells was 10 minutes (5 minutes before treatment and 5 minutes after treatment), the control cells were also acquired for a total of 10 minutes in order to ensure that photo bleaching was limited (as shown in S4 Fig) and therefore the effects of the treatment on the cell could be interpreted as resulting primarily from the treatment itself. Only the data of the first 5 minutes of the control cells are shown in the results section.”

Reviewer's comment 6.

When it comes to statistical analysis, I’m not sure that lumping all of the many time-resolved events together via an average makes the most convincing argument for differences. While it is great and important to show the very high temporal resolution of the method, it would be more interesting and probably more statistically powerful to look at the summation of events over time (on a per cell basis) as opposed to averaging over time. Other alternatives would be analyzing specific time points or the derivative/rate of change.

Authors' response 6.

This is an excellent point, to expand on the statistical analysis beyond the temporal resolution. We have now also added the plots of the cumulative sum of the events over time and included the average as well as the confidence interval between the experimental replicates at each time point in Fig 10B. 

 

Minor critiques:

Reviewer's comment 7.

Figure 6:

The control and post treatment time lapse curves appear identical. Please verify.

Authors' response 7.

We greatly appreciate that the reviewer picked up on this. We apologise for this and have now corrected it in the revised manuscript.

Reviewer's comment 8.

Secondly, the panels at the very bottom employ the use of a line graph that suggests a paired nature to the data. Bar graphs grouped by each event type would make the data easier to interpret to the reader. The error bars in the updated figure should be error between experimental replicates, not error between cells within an experimental replicate. Please add text to clarify that the number of events are per cell or per high powered field.

Authors' response 8.

We have replaced the interval and line graph with grouped bar graphs in Fig 6, as suggested. Furthermore, we have now removed the interval plots that were related to the error between cells within an experimental replicate in Fig 10.

Reviewer's comment 9.

Why are the control events so different from the pre-treated cells? It makes the reader wonder whether there is a high amount of biological variability (which could be addressed with repeat experiments) or a confounding variable has not been addressed in the experimental set up (temperature, CO2, pH?).

Authors' response 9.

We thank the reviewer for this comment. We have now chosen a micrograph that is more representative. 

Temperature, CO2 and pH have been tightly controlled (Tokai Hit system), similarly to all acquisition parameters. Low laser power was used in order to ensure photobleaching was limited (as validated in the control cells as shown in S4 Fig).

In addition, we indicate and refer to acquisition systems with fully automated fluidics system, which will be of high value for future high throughput screening and drug discovery, associated with cellular pathology with mitochondrial dysfunction, such as specific Parkin mutations and alike (line 553). 

Reviewer's comment 10.

Figure 7-9:

Why is there such a difference in the total number of detected structures in the last frame of untreated versus the first frame of the hydrogen peroxide-treated cells? The appearance of the cell has significantly changed suggesting that time 0 is not exactly the time at which peroxide was administered.

Authors' response 10.

The cell that was selected for Figs 8 and 9 to illustrate the pre-treatment and post-treatment groups unfortunately exhibited the behaviour that the reviewer pointed out. There was in fact some time that elapsed (5-10 seconds) between the end of the pre-treatment time lapse acquisition and the start of the post-treatment time-lapse acquisition. This is a limitation of the experimental setup. The difference in the total number of detected structures was however not consistently observed and we have therefore decided to change the representative cell for the pre-treatment and post-treatment conditions which does not exhibit this behaviour.

 

Reviewer's comment 11.

Furthermore, the overall cell morphology of the pre-treated and the control cell is completely different. I infer from the 2D representation that the pre-treated cell is more rounded and less adherent to the culture plate which is in contrast to the control which has the absence of mitochondria within the nuclear borders and spindle-shaped morphology and is adjacent to another cell. This suggests a different cell state which in and of itself can cause vastly different responses to treatment as well as baseline cell signaling characteristics. I would verify culture conditions are exactly the same between the groups. However, for representation of data, I would suggest picking cells that appear similar at baseline to show your reader.

Authors' response 11.

We thank the reviewer for this comment. We have now amended the pre- and post-treatment figures entirely and utilized a cell that appears similar in morphology (Figs 8 and 9).

Although our cell line was not synchronised, we now also refer to additional approaches that would minimize cell cycle dependent effects or would allow enhance control over cell geometry (line 528).

Reviewer's comment 12.

Discussion:

I would refrain from making statements about patterns observed that aren’t rooted in multiple observations with quantification across biological replicates with statistical testing of your hypothesis. In particular, making claims about sub cellular localization of events in reference to peri nuclear versus periphery when no distance data is presented in the paper may make the reader question your findings. Instead, I would suggest alluding how your method could be used to test those hypotheses in future studies.

Authors' response 12.

We have now edited the manuscript to remove specific reference to the subcellular localisation of events. In line 528 we now allude to the potential to investigate subcellular localisation of events in future studies. We also highlight the possibility to combine MEL with the calculation of other functional mitochondria parameters. 

Reviewer's comment 13.

Conclusion:

Other methods (including those you referenced) have achieved deterministic automated qualification of mitochondrial morphology, albeit in in 2D not 3D. I would qualify your closing statements so as to not discount the work of others.

Authors' response 13.

This is an important recommendation; we thank the reviewer for pointing this out. We have now edited this section in lines 534 and 553, so as to not discount previous work, but rather indicate the importance of building collectively upon the existing body of knowledge.

General paper edits.

We have supplied both the revised (clean) manuscript as well as a version indicating all amendments and edits.

---

## [Editor Report · Decision Letter 2]

15 Dec 2020

Mitochondrial event localiser (MEL) to quantitatively describe fission, fusion and depolarisation in the three-dimensional space

PONE-D-20-03864R2

Dear Dr. Theart,

We’re pleased to inform you that your manuscript has been judged scientifically suitable for publication and will be formally accepted for publication once it meets all outstanding technical requirements.

Kind regards,

Andreas S. Reichert, Dr. rer. nat.

Academic Editor

PLOS ONE
---

## [Editor Report · Acceptance letter]

17 Dec 2020

PONE-D-20-03864R2 

Mitochondrial event localiser (MEL) to quantitativelydescribe fission, fusion and depolarisation in thethree-dimensional space 

Dear Dr. Theart:

I'm pleased to inform you that your manuscript has been deemed suitable for publication in PLOS ONE. Congratulations! Your manuscript is now with our production department. 

Kind regards, 

on behalf of

Dr. Andreas S. Reichert 

Academic Editor

PLOS ONE